# The impact of inland ship emissions on air quality

Xiumei Zhang[1,2], Ronald van der A[1,2], Jieying Ding[2], Xin Zhang[1], Yan Yin[1]

[1]KNMI-NUIST Center for Atmospheric Composition, Nanjing University of Information Science and Technology (NUIST), Nanjing 210044, China

[2]Royal Netherlands Meteorological Institute (KNMI), Department of Satellite Observations, De Bilt, the Netherlands

*Correspondence to:* Yan Yin (yinyan@nuist.edu.cn) and Xiumei Zhang (xiumeizhang2019@outlook.com)

**Abstract.** Despite the large number of domestic inland river vessels in China, information on inland river vessel emissions is very limited, because limited legislation exists for emission control and there is no monitoring infrastructure. Taking the Yangtze River in the region of Nanjing as research area, we compiled a ship emission inventory based on real-time information

received from Automatic Identification System (AIS) signals and ship-related basic data provided by China Classification Society (CCS) database. The total estimated ship emissions in the Jiangsu section of the Yangtze River from September 2018 to August 2019 for $NO_x$, $SO_2$, $PM_{10}$ and $PM_{2.5}$ were 83.5, 41.8, 3.8 and 3.3 kton, respectively. These ship emissions were highest in the summer. From these estimates an inventory was constructed for ship emissions in the Yangtze River Delta (YRD) in Jiangsu. This ship emission inventory was compared with the Multi-resolution Emission Inventory for China (MEIC), the

Shipping Emission Inventory Model (SEIM) and the satellite-derived emissions using the Daily Emissions Constrained by Satellite Observations (DECSO) algorithm. The result shows a consistent spatial distribution with riverine cities having higher $NO_x$ pollution than non-riverine cities. With this comparison we analyzed the relative impact of ship emissions on densely populated regions along the river. Inland ship emissions of $NO_x$ and $SO_2$ are shown to contribute significantly, with each accounting for at least 40 %, to air pollution along the river.

**1 Introduction**

Maritime transport plays an essential role in national trade, and as the number of ships increases, so does the amount of emissions they emit into the air. The main pollutants emitted by ships are sulfur dioxide ($SO_2$), nitrogen oxides ($NO_x$ = NO +



$NO_2$), particulate matter (PM) and volatile organic compounds (VOCs) (Endresen, 2003; Moldanová et al., 2009; Tzannatos,

2010), all of which can affect regional air quality and human health (Capaldo et al., 1999; Dalsøren et al., 2009; Papanastasiou

and Melas, 2009; Eyring et al., 2010). In addition to direct pollution from ship emissions, the secondary generation of fine

particulate matter, sulfate and ozone further contributes to atmospheric pollution (Corbett, 1997). These pollutants are also

transported to large inland areas with sea and land breezes, seriously endangering human health and ecosystems (Dalsøren et

al., 2007; Endresen et al., 2007; Collins et al., 2009;  Eyring et al., 2010; Fan et al., 2016). Ship emissions have direct and

indirect effects on the radiation balance of the atmosphere. For example, $O_3$ chemically produced by $NO_x$ from ship emissions

can have positive radiative effects, while $SO_2$ emissions have a negative radiative effect and the sulfate produced by its

conversion also has a negative radiative effect at the Earth's surface (Endresen, 2003; Eyring et al., 2007, 2010; Lauer et al.,

2007).

According to the World Shipping Council (https://www.worldshipping.org/, last access: 21 November 2022), China possesses

7 of the world's top 10 biggest ports. The large number of ships in the ports have exacerbated air pollution in the local and

surrounding areas and increased traffic in the connected rivers. The data from the Ministry of Transport of China shows that

by the end of 2020, the number of inland river transport vessels is 11.50 million vessels, which is higher than the sum of coastal

transport and ocean transport in China (Statistical bulletin on the development of the transport sector in 2020). As one of the

most economically developed regions in the east of China, the Yangtze River Delta (YRD) region is the busiest inland river

ship transportation corridor in China. The Yangtze River is the third longest (6,300 km) river, and one of the busiest waterways

in the world. The YRD is the biggest port cluster  worldwide with as much ship emissions as the Bohai Bay and Pearl River

Delta port cluster together (Wan et al., 2020). Therefore, we focus on inland river ships in the Jiangsu section of the Yangtze

River as an important area for China to investigate the contribution of inland river ships to air pollution.

The methods of creating ship emission inventories have been developed and improved over the last decades, from the early

method of calculating emissions based on ship fuel consumption data (Streets, 1997; Corbett et al., 1999; Streets et al., 2000;

Endresen, 2003; Endresen et al., 2005, 2007; Psaraftis and Kontovas, 2009; Trozzi, 2010) to the method of calculating

emissions based on high temporal resolution ship navigation data provided by the Automatic Identification System (AIS).

Wang et al. (2008) pointed out that the regional emission inventories obtained from the fuel-consumption method are often





underestimated. For example, in North America, Europe and other regions, the values were only 20-70 % of the regional

inventory values established from the AIS method. Because AIS data include detailed information of real-time ship speed,

direction, location and many other parameters, it allows for accurate calculation of pollution emitted by ships. Jalkanen et al.

(2009, 2012) firstly developed the ship traffic emission assessment model (STEAM) using AIS system to study the effects of

ship emissions on regional air quality. Nowadays, the AIS method has been applied to many maritime regions, such as major

European seas, seas around Australia and other countries (Jalkanen et al., 2009, 2012, 2016; Kalli et al., 2013;; Goldsworthy

and Goldsworthy, 2015;Jonson et al., 2015; Johansson, 2017; Dragović et al., 2018).

Studies have also been conducted on ship emissions in Chinese ports such as the ports of Shanghai (Yang et al., 2007), Tianjin

(Chen et al., 2016), Qingdao (Liu et al., 2011), Hong Kong (Ng et al., 2013), Xiamen (Wang et al., 2020) and Shenzhen (Yang

et al., 2015). These studies were mainly based on fuel consumption methods. Nowadays, studies based on AIS methods to

establish ship inventories are gradually being carried out in China because the accuracy of this approach is higher than that for

the fuel-based method. More recently, studies in China are focusing on shipping emissions at coastal regions and ports (Song,

2014; Fan et al., 2016; Li et al., 2016; Huang et al., 2020) because the Ministry of Transport in China started the emission

control of ship emissions in 2016. Ship emission control zones were set-up in the waters of Beijing-Tianjin-Hebei, Yangtze

River Delta and Pearl River Delta to control, for example the sulfur content of the fuel. In addition, equipment with AIS is

becoming a mandatory requirement for ships of 100 gross ton and above. However, preliminary studies based on AIS methods

over inland river ship emissions showed a higher uncertainty than for coastal and ocean-going ships, especially for $SO_2$ and

$PM_{10}$ emissions (Li et al., 2016; Weng et al., 2020).

The total amount of air pollutants emitted by inland river vessels in the Yangtze River is higher than those emitted by seaport

vessels in Jiangsu province (Xu et al., 2019; Zhu et al., 2019). In contrast to the studies for ships on seas and in ports, studies

on inland river vessel emissions are still limited. Normally, those studies use estimations for the missing information about

ships, such as engine power and the maximum design speed. For example, Zhu et al. (2017) assumed that ship length is related

to ship tonnage and power. However, these methods still require a large amount of initial ship data, which is often difficult to

obtain. Moreover, the ship emission factors for the Jiangsu section of the Yangtze River used in the various studies show

significant differences. Their emission factors and the main engine fuel adjustment factors are usually based on the international



literature. Their applicability in China, as well as in Jiangsu province, needs to be validated. In addition, current studies of inland ship emissions have yet to explore the relative contribution of ship emissions to air pollution.

For the missing information on engine power in AIS data, our study proposes a method to reduce the difficulty in deriving information based on parameters directly provided by AIS. We set up an AIS receiver in Nanjing University of Information Science and Technology (NUIST) to collect ship information including ship position, speed and heading, ship name, ship length and ship type. Because of the large number of inland river ships in China and the limited range of antenna detection, we propose a method based on the length of the river per grid cell to extend the emission estimates to larger regions.

In this study, we compile a ship emission inventory of $NO_x$, $SO_2$, $PM_{10}$, and $PM_{2.5}$ for the Jiangsu section of the Yangtze River using a bottom-up approach based on AIS data from September 2018 to August 2019. We calculate the emission of each vessel at regular intervals (usually a few seconds) and sum up all emissions of the 588,591ships movements during one year. Emission characteristics such as ship type dependency, monthly variation and spatial distribution will be discussed. A comparison with the Multi-resolution Emission Inventory for China (MEIC) model, Shipping Emission Inventory Model (SEIM) and the Daily

Emissions Constrained by Satellite Observations (DECSO) method is performed to check the relative contribution of ship emissions to the total emissions. These results are important for the policy-makers to formulate and evaluate emission reduction policies and for ship companies in choosing the best emission reduction measures.

## 2 Methodology

### 2.1 AIS observations

The Automatic Identification System (AIS) is a transponder technology on board ships to enhance safety by broadcasting ship information via VHF (Very High Frequency) channels. It works in conjunction with the Global Positioning System (GPS) to broadcast information such as ship position, speed and heading, together with static ship information such as ship name, ship length and ship type. Each AIS-equipped vessel is identified by a unique Maritime Mobile Service Identity (MMSI) number, which is also part of the AIS data.



We have set up an observation location in Nanjing, where an AIS receiver is located on the roof of the meteorological building

in NUIST. Ships transmit AIS signals at intervals varying from every 3 seconds to a few minutes to provide information on

their position. The black dots in Fig. 1c are an example of the locations of ships according to the AIS signals received in a

time interval of 4 hours. It shows that the antenna can receive signals within a 50 km radius. A region with a longitude of less

than 118.95° E and a latitude of more than 32.05° N was selected as the area where all ships, including those with weak AIS

transmitter can be tracked under all conditions. AIS data were collected for 365 days from September 2018 to August 2019.

Based on the AIS information, ships were classified into seven types: cargo ships, tankers, passenger ships, tugboats, dredgers,

patrol vessels, and others. Cargo ships include container ships, bulk carriers, Ro-Ro (Roll-on/Roll-off) vessels (excluding

passenger ships); tankers comprise liquid chemical tankers, liquefied gas tankers and oil tankers; passenger ships consist of

ferries and Ro-Ro passenger ships. Patrol vessels are classified separately because of their small size in combination with high

speed.





**Figure 1. The map of the study area. (a) Map of the Yangtze River. (b) Schematic map of the Jiangsu section of the Yangtze River. (c) Ship locations that were received by AIS in a 4 hour interval shown on a map with a resolution of 0.1°×0.1°. The black dots are the ship locations received by AIS, which demonstrates the range of the AIS receiver. The blue box is the selected observation area with a longitude between 118.65 ° E and 118.95° E and latitude between 32.05° N and 32.25° N.**






The AIS data show that approximately 2,000 boats pass through the observation area each day, with a drop in the number of ships on specific days. Figure 2 shows the number of unique ships per day over a year. The grey box in Fig. 2 shows the Chinese New Year period when the number of vessels dropped sharply. The decrease in the number of ships on other days is usually related to the weather conditions. On 26 November 2018, very thick fog occurred, and the travel of both ships and

vehicles was disrupted. On 24 February 2019, the local meteorological bureau issued a fog warning, so the number of passing vessels decreased. On 10 August 2019, Typhoon Lekima made landfall and brought catastrophic damage to Southeast China. The number of ships coming from the typhoon region was greatly reduced, as was the number of local vessels in Nanjing. On 5 January 2019, the Jiangsu Maritime Bureau issued a navigational warning for construction work in the Baguazhou branch of the Yangtze River in Nanjing.

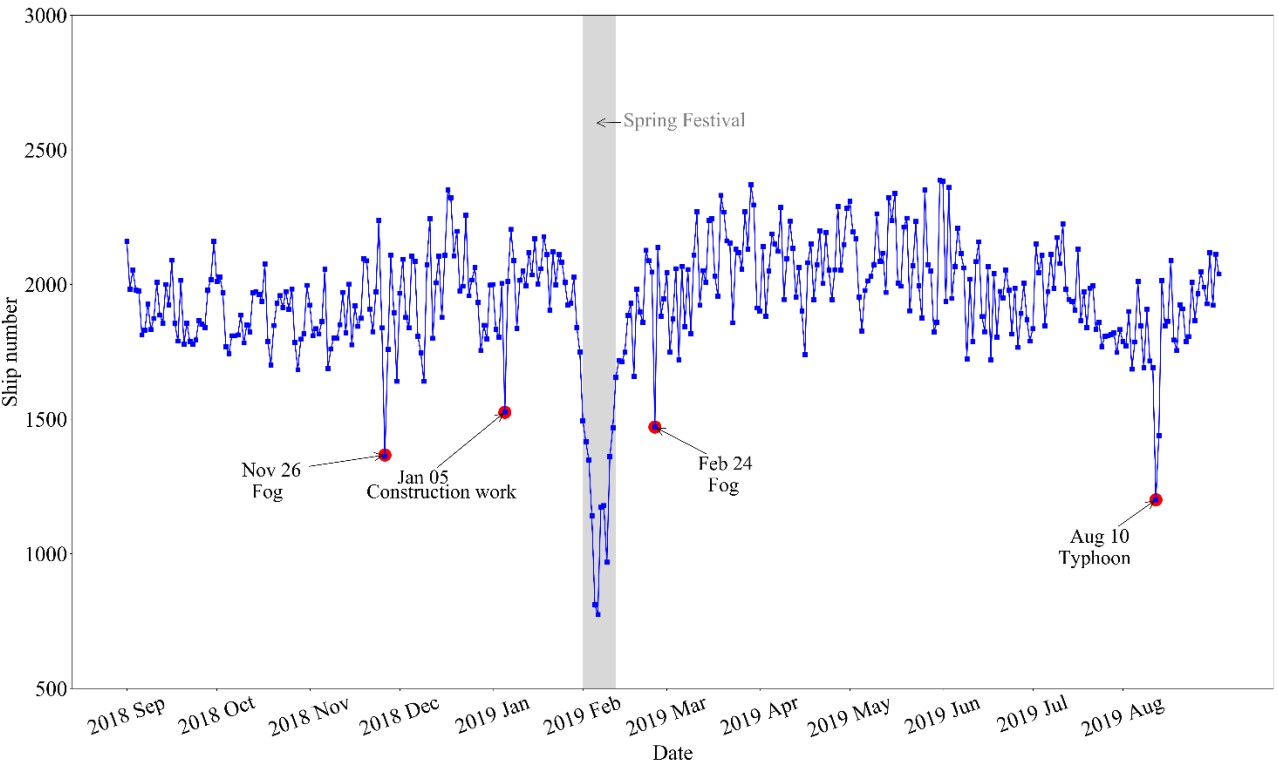


**Figure 2. Number of ships per day. The grey shaded area represents the Spring Festival period. The red dots indicate days with largelyreduced ship numbers.**



## 2.2 Ship emission estimation

To estimate shipping emissions from the Yangtze River, we adopt the AIS method to obtain high-resolution ship information

and emission factors. Equation (1) was used for calculating the emissions of various pollutants such as $NO_x$, $SO_2$, $PM_{2.5}$, and

$PM_{10}$ from a single ship based on its main power (Jalkanen et al., 2012). We only considered the emissions of the main engine

of the ship, while ignoring the relatively lower emissions from the auxiliary engines.

$$E = P \cdot L \cdot f_{EF} \cdot f_{LLAM} \cdot f_F \cdot f_C \cdot T \qquad , \tag{1}$$

where

130        $E$ is the emission from the main ship engine;

           $P$ is the main engine power of the ship (See Section 2.2.1);

           $L$ is the main engine load factor (See Section 2.2.2);

           $f_{EF}$ is the emission factor;

           $f_{LLAM}$ is the low load adjustment multiplier of the main engine;

135        $f_F$ is the fuel correction factor;

           $f_C$ is the control factors for implementation of greener technology;

           $T$ is the sailing time.

Below we will further discuss each element of this equation.

The emission factor ($f_{EF}$) refers to the mass of pollutants emitted per unit of work done by a single vessel in one hour (g kW$^{-1}$

h$^{-1}$). It is an essential parameter for calculating ship emissions and has not yet been well studied in China. For example, the

emission factors used by Wan et al. (2020) for Beijing-Tianjin-Hefei, Yangtze River Delta, and Pearl River Delta were taken

from literature referred to Europe and the US (USEPA, 2000; Entec, 2002; EPA, 2009). Ship emission factors used by Fan et

al. (2016) for the Yangtze River Delta and the East China Sea were based on an EPA report (2009) and on Goldsworthy and

Goldsworthy (2015), while the latter is a study for the Australian region. The emission factors used by Fu et al. (2012) for

Shanghai port were taken from the air pollutant emission inventory of the Port of Los Angeles (Archana et al., 2013). Lacking

the studies for ships of the Chinese fleet, our study also relies on emission factors for other regions, following the scheme in

Table 1.





The low load adjustment factor $(f_{LLAM})$ is used when the main engine load factor is less than 0.2. When the load factor is below 0.2, there is an increase in emission intensity due to the inefficient engine use at low speed. Hence, the emissions from low-load vessels were multiplied by the low-load adjustment factor (EPA, 2009). The low load adjustment multipliers for the main engine are shown in Table S1.

The fuel correction factor $(f_F)$ is correcting the emission factors according to sulfur content of the fuel used by the ships. The emission factors mentioned above are obtained based on the assumption that marine fuel is heavy oil with a sulfur content of 2.7 % (EPA, 2009). Based on the fuel monitoring data from the Nanjing Maritime Bureau, Xu et al. (2019) found that the sulfur content of inland river vessel fuel varies from 0. 19 to 2 %, with a medium value of around 1.5 %. In our study we use the average sulfur content of marine distillates (MD) of 1.5 %. The fuel correction factor for each air pollutant is shown in Table 2.

The control factor $f_C$ is to potentially correct for higher quality marine diesel engines, or the availability of emission reduction facilities on the ships as a result of emission control policy. There is currently limited legislation to control ship emissions in inland waterway zones in China and insufficient monitoring infrastructure for enforcement, therefore the control factor is set to 1.

**Table 1. Emission factor $(f_{EF})$ in g $(kWh)^{-1}$ of the main engine reported in literature. (SSD is slow speed diesel; MSD is medium speed diesel. ≤1999, 2000-20102, 2011-2015 are the years of the ship built.)**

| Engine type | $NO_x$ | $SO_2$ | $PM_{2.5}$ | $PM_{10}$ | Reference |
|---|---|---|---|---|---|
| SSD | ≤1999:18.1<br>2000-2010:17.0<br>2011-2015:15.3 | 10.5 | 0.96 | 1.05 | (Entec, 2002) |
| | 17.0 | 10.5 | 1.2 | 1.5 | (Archana et al., 2013) |
| | 18.1 | 10.3 | 1.22 | 1.378 | (EPA, 2009) |
| MSD | ≤1999:14.0<br>2000-2010:13.0<br>2011-2015:11.2 | 11.5 | 1.02 | 1.11 | (Entec, 2002) |
| | 13.0 | 11.5 | 1.2 | 1.5 | (Archana et al., 2013) |




| | 14.0 | 11.31 | 1.22 | 1.193 | (EPA, 2009) |
| HSD | 12.7 | 11.31 | 0.50 | 0.650 | (EPA, 2009) |


**Table 2. Fuel correction factor ($f_F$) used in this study for a sulfur content of 1.5 %**

| Fuel type | Sulfur content (%) | NO$_x$ | SO$_2$ | PM$_{2.5}$ | PM$_{10}$ |
|---|---|---|---|---|---|
| MD | 1.5 % | 0.9 | 0.56 | 0.47 | 0.47 |

### 2.2.1 Engine power

Since the engine power is missing in the AIS data, we develop a method to relate the engine power to the ship type, length and

speed. Those parameters are available in the AIS data.

The supplied power by ship engines ($P_{ship}$) for transport should be at least equal to the dragging force ($F_{drag}$) of the water

resistance multiplied by the speed ($v$) of the ship. According to fluid dynamics, the resistance force is proportional to the area

($A_s$) experiencing the water resistance multiplied by the speed square. Hence, the supplied power of a ship engine is

proportional to the surface area times the cubic of the speed: $P_{ship} \sim A_s v^3$. This is known as the propeller law (Theotokatos and

Tzelepis, 2015).

For each ship type, the length-width-height dimension ratios of the boats are not expected to vary much, especially because

ship design is focused on efficiency as fuel is a dominant cost in shipping. Logically, ship designs are expected to be relatively

similar. Hence, the ship area $A_s$ (width times height) is to the first order expected to be proportional to the square of the ship

length ($A_s \sim L_s^2$). Therefore, we assume the relation of engine power of a ship with its length and its speed as: $P \sim L_s^2 v^3$. We

derive the exact relationship for each category of ship by linear fitting of this proxy using the ship parameters registered in the

China Classification Society (CCS, https://www.ccs.org.cn/ccswz/, last access: 5 December 2022) database of Chinese

domestic ships. The fitted linear relation between $L_s^2 v^3$ and the engine power $P$ of the main engine is shown in Table 3 for each

ship category. Figure S2 shows fitted linear relationship between main engine power at full speed and the proxy $L_s^2 v^3$ at

maximum speed $v_{max}$.






**Table 3. Regression analysis between main engine power and the square of the vessel's length multiplied by the cube of the real-time speed (The slope refers to the ratio between power ($P_{max}$) and $L^2 v_{max}^3$). When the tug's length is more than 40 meters, the power of the main engine is assumed to be constant (1800 kW).**

| Type | | Slope (Mg m⁻³) | R² | Sample size |
|---|---|---|---|---|
| Cargo | | $4.755 \cdot 10^{-5}$ | 0.87 | 744 |
| Tanker | | $8.692 \cdot 10^{-5}$ | 0.91 | 487 |
| Passenger | | $4.260 \cdot 10^{-5}$ | 0.88 | 226 |
| Tug | $L_s \leq 40m$ | $5.408 \cdot 10^{-4}$ | 0.96 | 213 |
| Dredger | | $1.258 \cdot 10^{-4}$ | 0.94 | 10 |
| Patrol | | $4.869 \cdot 10^{-5}$ | 0.93 | 37 |
| Others | | $6.906 \cdot 10^{-5}$ | 0.85 | 184 |

**2.2.2 Load factors**

The load factor $L$ reflects the actual output power of the main engine as a percentage of the rated main engine power. The ship's main engine works according to the characteristics of a propeller, and the engine load factor is estimated using the propeller law, which means that the main engine load varies as the third power of the ratio of the ship's actual speed to the ship's designed speed. So the load factor is calculated as:

$L = (\frac{v}{v_{max}})^3$  ,                                                                                     (2)

where $v$ is the actual sailing speed of the ship; $v_{max}$ is the maximum speed of the ship using its full power.

For inland waterway ships, the maximum designed speed is often unknown and cannot be obtained directly from AIS data. To obtain a maximum sailing speed per ship type we used the median of all maximum speeds (per selected ship type) registered in the CCS ship database (see Table 4).


**Table 4. Maximum speed per ship category (The maximum speed value is based on the statistics of the maximum speed of about 1,900 different river vessels.)**





|  | Cargo | Tanker | Passenger | Tug | Dredger | Patrol | Others |
|---|---|---|---|---|---|---|---|
| $v_{max}$ (km h$^{-1}$) | 15.33 | 12.00 | 14.30 | 13.35 | 10.76 | 19.80 | 12.80 |

### 2.2.3 Correction of ship velocity for river flow

From the previous equations for calculating ship emissions, we see that the ship's real-time speed is critical. The speed related to the delivered power by the ship is the speed relative to the water, rather than the ship speed over ground, which is given in the AIS signal. Therefore, a correction for the vessel speed is required.

Assuming that the ships use similar engine power going upriver or down-river, the average speed difference between ships going downstream and upstream has been used to derive the speed of the water flow. Then the actual speed of a ship is obtained

by correcting the speed with the actual river flow. When the obtained speed for an individual ship is negative, this ship is assumed to be at anchor and the sailing speed of the ship is set to 0. Figure 3 shows the daily average speed of the river flow during a year. It is clear from Fig. 3 that the water speed is higher in summer than in winter, because of the difference in rain and melting water over the year. This is confirmed by a hydrological study by Chen et al. (2016), concluding that monthly precipitation and discharge in the Yangtze River basin are highest in July and August, with January, February and December

being the lowest months. The quick change in river flow on August 10, 2019, happened during the landfall of typhoon Leikma. The actual ship speed after correction for water speed tends to be rather constant throughout the year. The actual engine power is calculated using the corrected speed as discussed above.

Note the limitations of our method. Some ships would reduce their speed when going downstream, or some increase their speed when going upstream to compensate for the engine output. In this case, our method may slightly underestimate the river's

speed, especially in July and August.





**Figure 3. Average daily river speed (red line) and ship speed over a year. Up-river speed and down-river speed of the ships are coming directly from AIS data. The river speed is based on the average difference between downstream speed and upstream speed.**

**2.3 Emission inventories for the Yangtze River Delta**

To evaluate the emissions derived using our method, we compared results with three emission inventories in the YRD, including the satellite-derived $NO_x$ inventory DECSO (Daily Emission derived Constrained by Satellite Observations) and the following bottom-up inventories for East Asia: MEIC (Multi-resolution Emission Inventory for China) and SEIM (Shipping Emission Inventory Model).

### 2.3.1 MEIC

The Multi-resolution Emission Inventory for China (MEIC, http://meicmodel.org/, last access: 5 December 2022), developed by Tsinghua University, is an emission inventory of air pollutants from anthropogenic sources in China with a spatial resolution of 0.25°. Emissions of $NO_x$, $SO_2$, $PM_{10}$, $PM_{2.5}$, $CO_2$, OC and BC are calculated for four sectors from 2008 to 2017: energy, industry, transport and residential. For road motor vehicle emission sources, MEIC uses an emission characterization model that includes parameters such as temperature, humidity and combines meteorological fields with motor vehicle emission factor models to construct a high-resolution dynamic motor vehicle emission inventory. However, the transport sector of MEIC does not include ship $NO_x$ emissions (Zheng et al., 2014). Zheng et al. (2018) and Li et al. (2017) presented more details of the latest version MEIC v1.3. Here, the MEIC v1.3 for 2017 is used.

### 2.3.2 SEIM

The Shipping Emission Inventory Model (SEIM) developed by Tsinghua University has been used to construct the East Asia ship emission inventory. The model has been developed based on high-precision AIS information of ocean-going ships and encompassing worldwide international fleet activity. It provides gridded annual ship emission data with a 0.1° spatial resolution for the seas in the East Asia region in 2017, covering $SO_2$, $NO_x$, CO, VOC, $PM_{2.5}$, OC and BC, a total of seven species (Liu et al., 2019). However, emissions over inland rivers, except for the delta region, are not included.

### 2.3.3 DECSO

Daily Emission estimates Constrained by Satellite Observations (DECSO) is an inverse modeling method to update daily emissions of $NO_x$ based on an extended Kalman filter (Mijling and van der A, 2012). $NO_x$ emissions are constrained by combining simulated $NO_2$ column concentrations of a regional chemical transport model (CTM) with satellite observations. The latest version is referred to as DECSO v6.1 and has a spatial resolution of 0.1° using $NO_2$ observations from TROPOMI (TROPOspheric Monitoring Instrument). In addition, the algorithm captures the seasonality of $NO_x$ emissions and reveals the trajectory of ships near the Chinese coast (Ding et al., 2018). DECSO provides monthly emissions with a maximum error of

approximately 20 % for each grid cell. We selected $NO_x$ emissions in the Yangtze River Delta region for 2019, the earliest year that is available from DECSO for this region.

## 3 Inland ship emissions

We calculated the emissions of $NO_x$, $SO_2$, $PM_{10}$, and $PM_{2.5}$ for the area considered (118.65°-118.95° E and 32.05°-32.25° N, the purple box in Fig. 1) from September 2018 to August 2019 based on AIS data. The total ship emissions of $NO_x$, $SO_2$, $PM_{10}$ and $PM_{2.5}$ in this area are 6,679 and 3,341, 292 and 265 ton per year, respectively.

### 3.1 Yearly emissions per ship category

We analyzed the collected AIS data from September 2018 to August 2019 and concluded that the number of ships for each

defined type was stable throughout the time period. Detailed data on the number of vessels per month are listed in Table S1. Figure 4 shows the contribution of $NO_x$, $SO_2$, $PM_{10}$ and $PM_{2.5}$ from different ship types. Emissions from cargo ships are higher than other types of ships, followed by tankers and tugs, with dredgers and patrol boats making the lowest contribution to the emissions. Cargo ships contribute to more than 58 % of the ship emissions of all species in the Nanjing section of the Yangtze River. This is because cargo ships are the dominant vessel type in this region and their number account for about 81 % of the

total.

Dividing the total emissions from each ship type by the number of ships of that type we obtained and analyzed average ship emissions per ship type. For a single vessel, tugs emit more pollutants (Fig. S2). This is related to the higher power needed for the engine of a tugboat, which was also concluded by Xu and Bai (2017). Compared with cargo ships, tankers have a relatively higher contribution to $SO_2$. The diesel sulfur content of inland river cargo ships is about 0.19-2 %, which is higher than the

emission control zone requirements of 0.5 % (Zhu et al., 2019). The sulfur content of tanker fuel is around 1.0-3.5 % (Zhu et al., 2019). Because $SO_2$ emissions are directly related to the sulfur content of fuel oil, tankers will emit more $SO_2$.



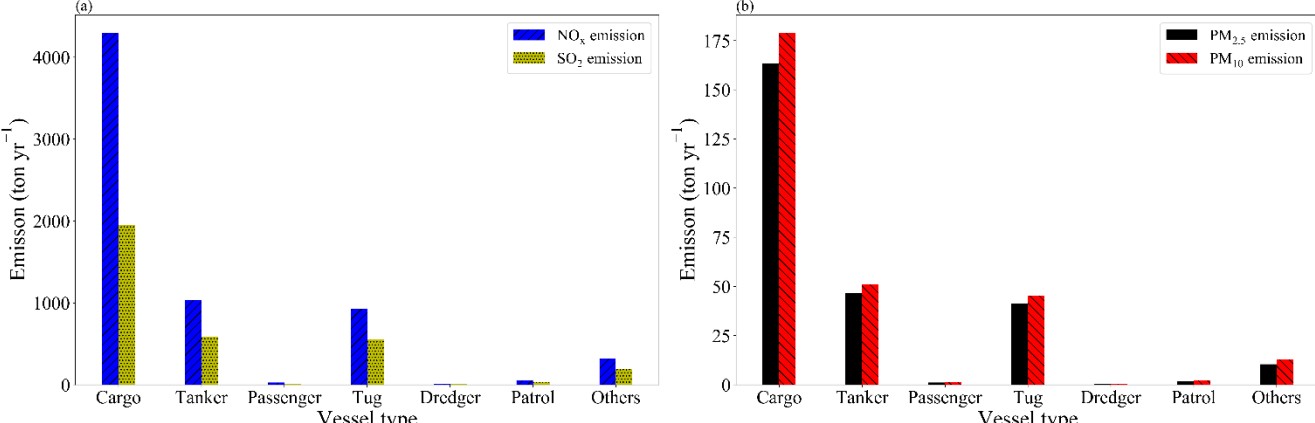

**Figure 4. Emissions from different ship categories**

**Table 5. Number of ships and NO$_x$ emissions for the various ship categories**

|  | Ship number | Share | Rank | NO$_x$ emission (kton yr$^{-1}$) | Share | Rank |
|---|---|---|---|---|---|---|
| Cargo | 571013 | 81.0 % | 1 | 4.3 | 64.4 % | 1 |
| Tanker | 96888 | 13.7 % | 2 | 1.0 | 15.5 % | 2 |
| Tug | 12128 | 1.7 % | 3 | 0.9 | 13.9 % | 3 |
| Others | 10932 | 1.5 % | 4 | 0.3 | 4.8 % | 4 |
| Passenger | 7971 | 1.1 % | 5 | 0.027 | 0.41 % | 6 |
| Patrol | 5652 | 0.8 % | 6 | 0.057 | 0.86 % | 5 |
| Dredger | 346 | 0.05 % | 7 | 0.001 | 0.16 % | 7 |




### 3.2 Monthly variation of ship emissions in the observational area

We calculated the monthly emissions from the AIS data in the observational area, as is shown in Table 6. Since the same calculation method is adopted for all four pollutants, their spatial and temporal variations are consistent. Figure 5 shows the monthly variations of $NO_x$ emissions from ships from September 2018 to August 2019. We see that, except for the sharp decrease in the number of ships in February, the number in other months has basically stabilized at around 60,000. The monthly emissions are highly related to the number of ships and the ship speed.

Ship emissions peaked in July, followed by June and August. In July the emissions are the highest due to the high river speed and therefore more engine power (proportional to $v^3$) is needed for upstream ships. From November to February, the emissions have been lower, possibly because the water flow of the Yangtze River is lower at this period and ship activities during Spring Festival are reduced. In February, the number of ships dropped sharply, but the emissions from ships in February were similar to that from ships in January. This is due to the higher activity of tugs in February, which have larger engine and emit higher levels of emissions (Fig. S3). From February to March, we see that pollutant emissions increased due to the increase in the number of ships. This is closely related to the resumption of factory work and human activities after the holiday.

**Table 6. Ship emissions (ton yr$^{-1}$) in our study region around Nanjing from September 2018 to August 2019**

| Month | $NO_x$ | $SO_2$ | $PM_{10}$ | $PM_{2.5}$ |
|---|---|---|---|---|
| Jan | 465 | 236 | 20 | 18 |
| Feb | 452 | 237 | 20 | 18 |
| Mar | 590 | 298 | 26 | 24 |
| Apr | 575 | 296 | 25 | 23 |
| May | 634 | 321 | 28 | 26 |
| Jun | 694 | 340 | 31 | 28 |
| Jul | 787 | 371 | 36 | 33 |
| Aug | 680 | 340 | 30 | 27 |
| Sep | 497 | 248 | 21 | 19 |
| Oct | 467 | 235 | 20 | 18 |





| | | | | |
|---|---|---|---|---|
| Nov | 406 | 204 | 17 | 16 |
| Dec | 432 | 217 | 18 | 17 |

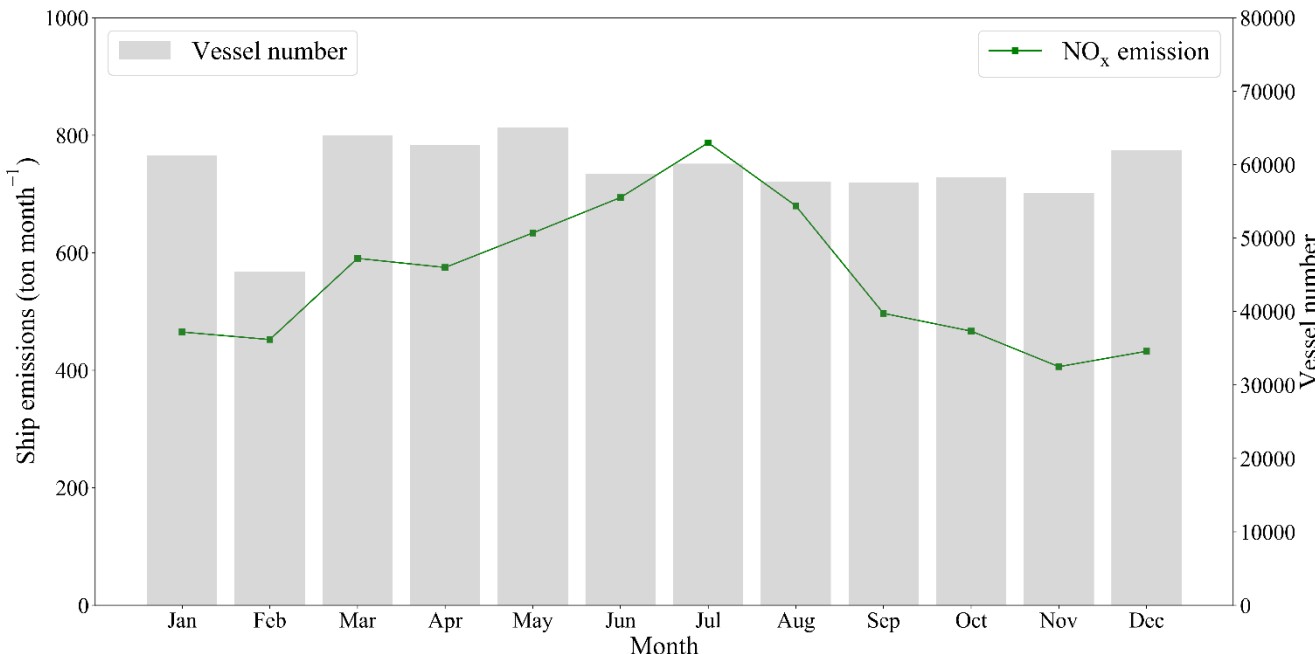

**Figure 5. Monthly changes in the emissions from inland river ships. The grey bars show the number of ships per month.**

## 3.3 Spatial distribution of NO$_x$ emissions

### 3.3.1 Spatial distribution of NO$_x$ emissions in the observational area

The spatial distributions of emissions from different air pollutants are the same as they are based on the shape of the river. Here we choose NO$_x$ as an example to present our results. Figure 6 shows the calculated ship emissions with a spatial resolution of 0.01° in the observational area, not only over the main channel, but also over some very small branches of the Yangtze River, such as the Chu River and Ma Cha River. The observational area also includes a slightly wider branch of the Yangtze River, known as the Jia Jiang River. On these branches, ship emissions are much smaller than from ships on the main Yangtze River. 300 The Yangtze River is around 2-3 km wide.

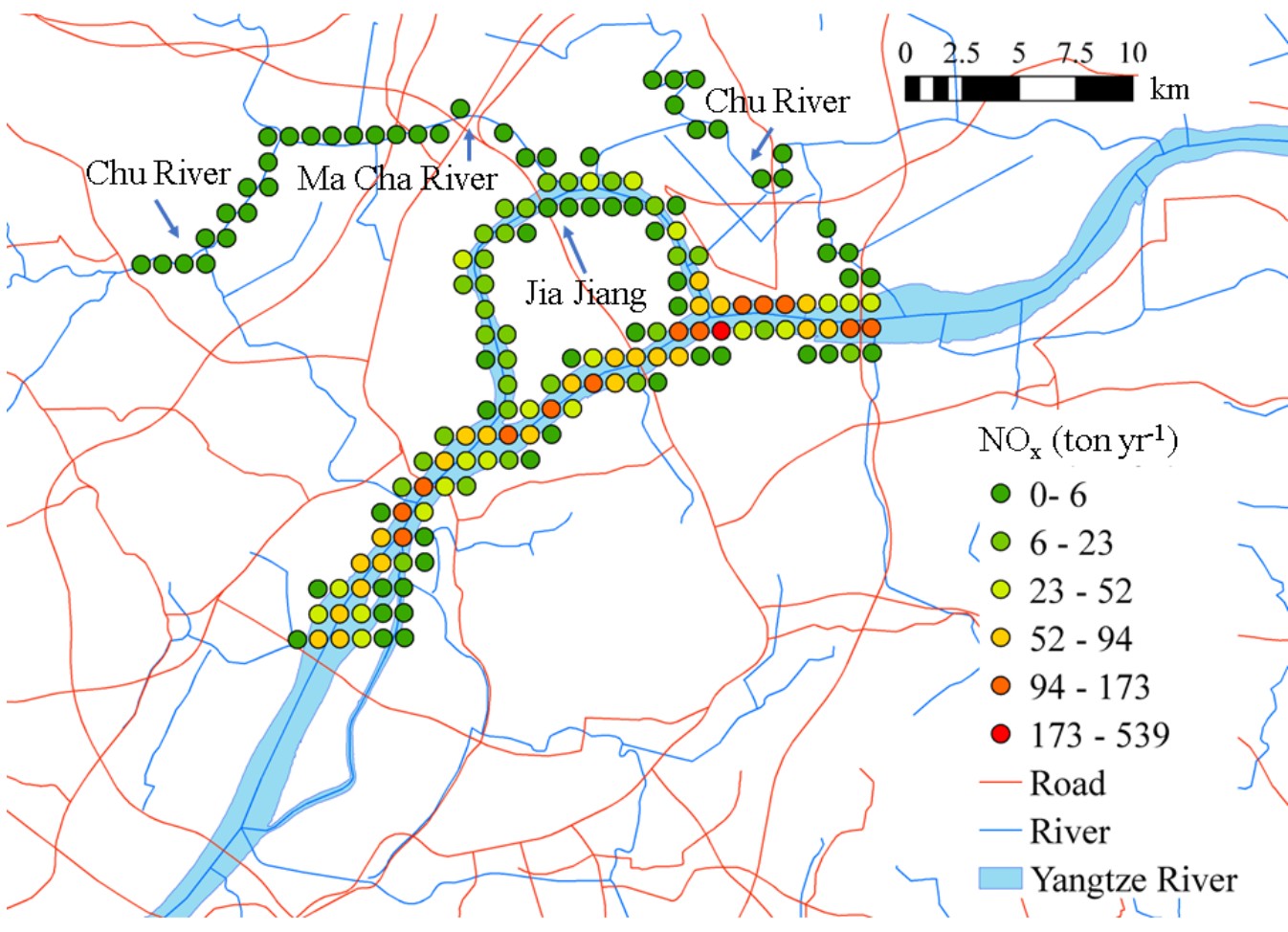

**Figure 6. Spatial distribution of ship emission in the observation area on a resolution of 0.01° × 0.01°**

### 3.3.2 Spatial distribution of NOx emissions in the YRD region

Due to the limits of the observation range, we propose a new river length-based method to estimate ship pollution in a larger

region. Emissions in inventories are often presented per grid cell. Figure 1 shows that the river width is much smaller than the

size of a grid cell and emissions from ships are more closely related to river length per grid cell than river width. Therefore,

inland river ship emissions can be seen as line sources, assigning pollution to each kilometer of the river. First, we calculate

the emissions per km length of the river. Once we have this quantity, we can calculate the ship emissions per grid cell for any

part of the river based on the length of the river in that particular grid cell. For parts of the Yangtze River having parallel

branches, the ship emissions are distributed by the corresponding proportion of the number of ships per branch.

We allocated the total ship emissions $E_{\text{all}}$ in the observational area to the length $L_{\text{river}}$ of the mainstream river only: $E_{\text{all}}/L_{\text{river}}$.

The length of the main channel of the observational area can be inferred by analyzing the distance traveled by ships in

downstream and upstream directions (Fig. S4) using the location, speed and time of the ships from the AIS data. We only

calculated the distance for vessels that have fully passed through the mainstream channel with a speed of more than 3 knots

downstream and more than 1 knot upstream for the entire journey. The downstream side of the river is approximately 30 km

and the upstream side is around 32 km. The average length of the main channel was calculated to be 30.9 km, weighted by

ships navigating in both directions. We can now obtain the ship emissions within each grid cell by multiplying the river's length

within the grid cell by the calculated emissions per kilometer of the  river: $E_{\text{all}}/L_{\text{river}}$ (include the value (kton yr$^{-1}$ km$^{-1}$)). The

Jia Jiang River has been ignored in this calculation since only 0.04 % of all ships pass this side river.

According to the above method, we extrapolate our emissions per km to the river outside our study area assuming that the ship

density and speed is rather constant in this region. Figure 7 shows the spatial distribution of ship emissions in the Jiangsu

section of the Yangtze River with a spatial resolution of 0.1 degree. The spatial distribution of our ship emissions depends only

on the length of the river within the grid cell. On average, 17 % of the ships in the observational area are at the dock every day,

and this part of the ship emissions have not taken into account. The auxiliary engines are still working when the ship is moored,

and according to the US EPA (EPA, 2009), the ratio between the ship's auxiliary and main engines is about 20-25 %. Thus, we

underestimate the emissions by 3 to 4 %.

However, the locations of high ship emissions are consistent with previous studies. Zhu et al. (2019) pointed out that the

distribution of ship emissions in the Jiangsu section of the Yangtze River in 2017 was uneven, with the emission rates in the

Nanjing section of the Yangtze River and the Jiangyin section of the Yangtze River being relatively high. Xu et al. (2019) noted

that for ports along the river, Nanjing port had the highest rate of ship emissions.

As the Yangtze River becomes wider when getting closer to the sea, the speed of the river will be reduced and thus the emissions

from ships can be lower. In the extreme cases of stagnant water, ship emissions can be reduced by a maximum of 3-33 %



depending on the month. We estimate that this may lead to an error of about 10 % in the ship emissions outside our study area

around Nanjing.

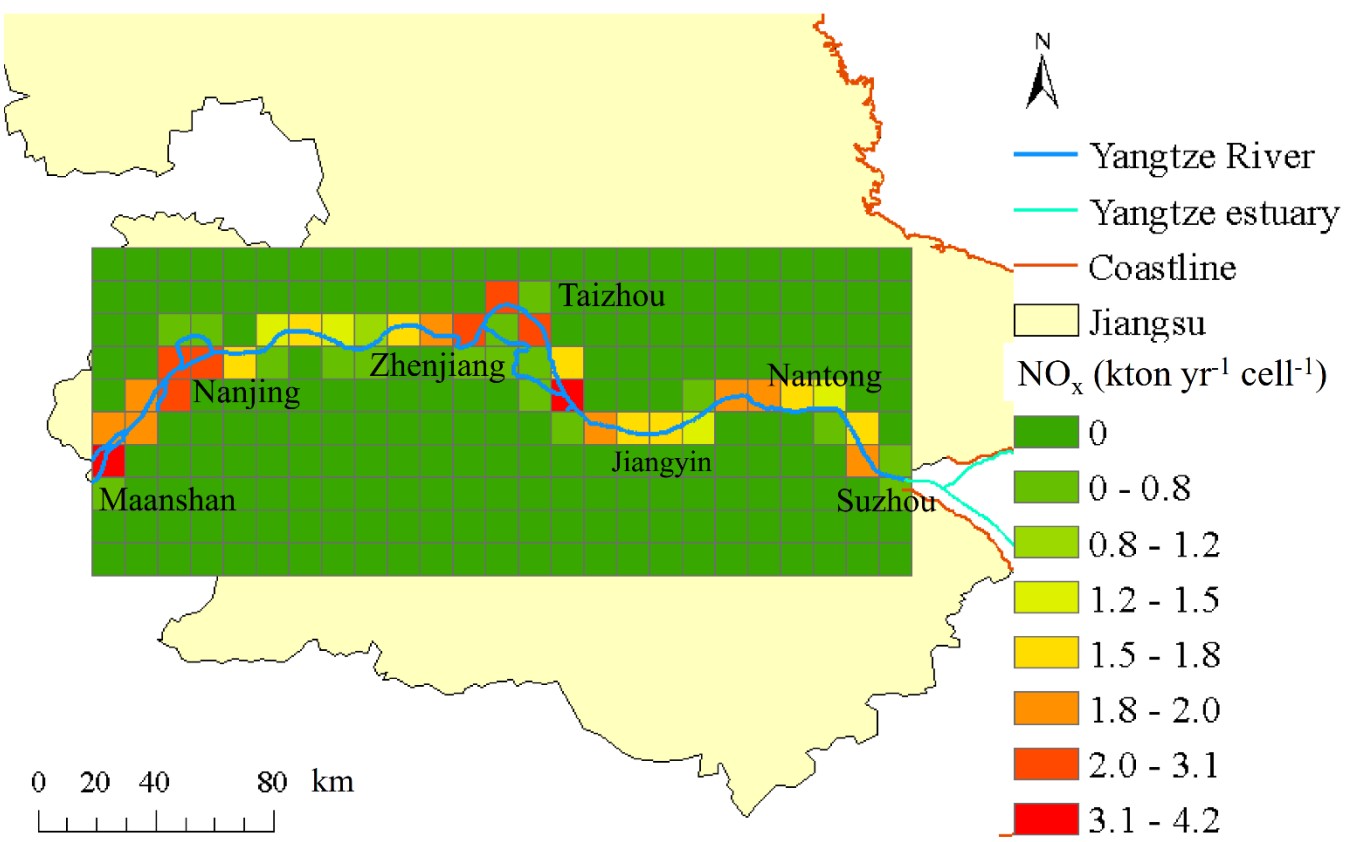

**Figure 7. Spatial distribution of NOₓ emissions in the Jiangsu section of the Yangtze River**

**3.4 Uncertainty**

Besides the uncertainties of the emissions factors, two of our assumptions may also bring certain uncertainty in estimation of

emissions. Firstly, we did not account for emissions from moored vessels, assuming their emissions are negligible. By

calculating the non-moving vessels in the AIS data and using emission data from the literature for auxiliary engines, we

estimated this could lead to a maximum underestimation of 4 %. Secondly, our method relies on the actual ship speeds, which

we assumed to be rather constant over the Yangtze River. However, in sections of the Yangtze River that are wider and having more water volume, especially close to the sea, the river speed is likely to be lower. It can be calculated that emissions from

ships in these areas can be lower by maximum 12 % in the extreme case of zero river speed. Taken these assumptions into account and the uncertainties on the emission factors, we estimate a total uncertainty of ± 15 %.

## 4 Importance of inland ship emissions

We calculated our ship emission inventory called JSEI over the Jiangsu section of the Yangtze River (118.5°-121° E and 31.5°-32.5° N) and compared it with other regional inventories. In the following discussions, we only selected the emissions from

the grid cells containing the Yangtze River and the contribution of ship emissions to the total emissions of air pollutants along the river region was analyzed.

### 4.1 A comparison of the ship emission inventories

To verify the calculated emissions of JSEI, we compared them with the sea-going ship emissions of SEIM over the overlapping region. Figure 8 shows the spatial distribution of the ship emissions of $NO_x$, $SO_2$ and $PM_{2.5}$ for JSEI and SEIM and the ratio

of JSEI to SEIM. JSEI and SEIM only calculate ship emissions, and the two overlap for less than half of domain. When comparing the overlapping grid cells, JSEI accounts for in average about 99 %, 59 % and 40 % of the SEIM emissions for $NO_x$, $SO_2$ and $PM_{2.5}$, respectively. The average emissions from inland ships over rivers (JSEI) compare well with average emissions of sea-going ships (SEIM) for $NO_x$. SEIM has higher values than JSEI for $SO_2$ and especially for $PM_{2.5}$. Inland river emissions are still high, and further regularization of inland ship emissions is needed.



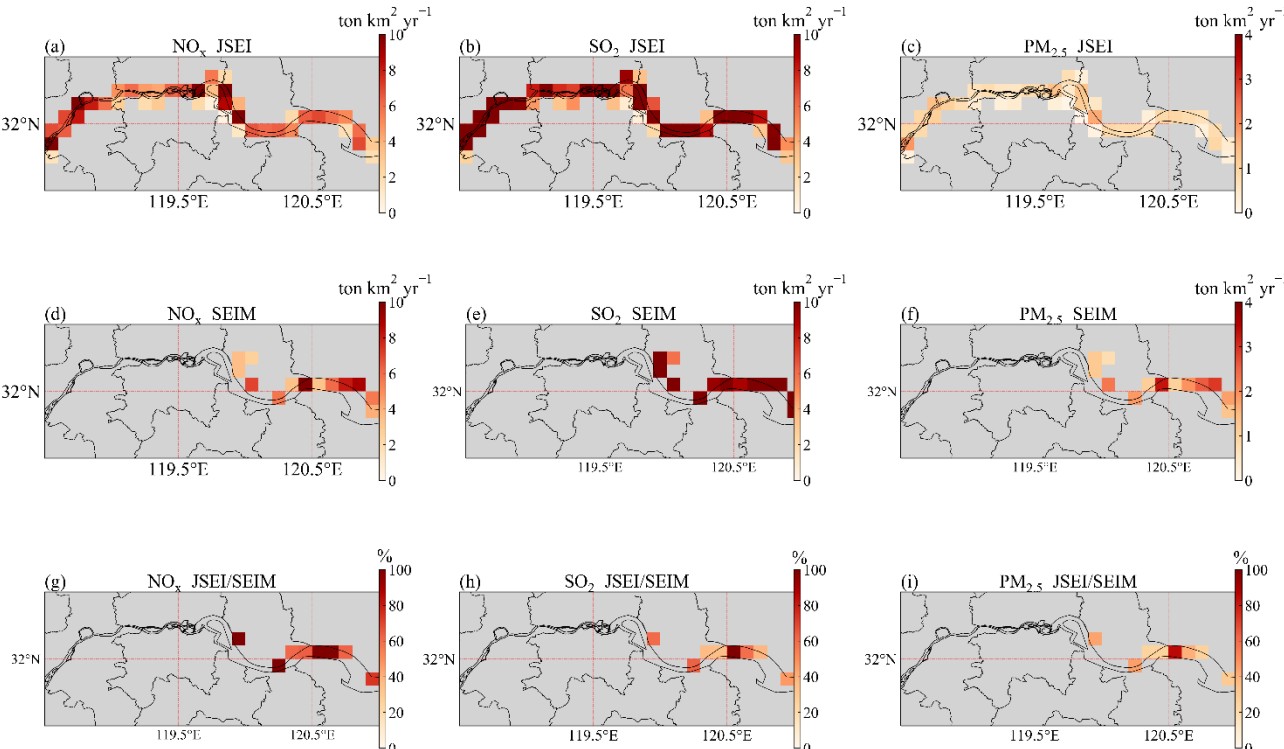


**Figure 8. Spatial distribution of ship emissions for JSEI and SEIM.**

**4.2 Contribution of ship emissions compared to MEIC and DECSO**

Here JSEI is added to the MEIC inventory to represent the value of total air pollutants, as the MEIC inventory does not include

emissions from ships. Figure 9 shows the spatial distribution of the contribution of the river emissions JSEI over the total

emissions (MEIC+JSEI) for $NO_x$, $SO_2$, $PM_{2.5}$ and $PM_{10}$ along the river. PM emissions from ships account for a relatively low

proportion of total PM, with a maximum of 20 % and an average of 5 % in riverine areas, with $PM_{2.5}$ accounting for a slightly

higher proportion than $PM_{10}$. $SO_2$ emissions from ships contribute to around 40 % of the total $SO_2$ pollution on average, while

it can reach 83 % in some areas. Therefore, it is important to set up control zones for sulfur emissions from ships.

We integrated JSEI to 0.1° resolution and 0.25° resolution to compare with DECSO and MEIC, respectively. Because the

DECSO only covers $NO_x$ emissions, we focus on the contribution of $NO_x$ emissions from ships to the total emissions. Figure




10 shows the spatial distribution of $NO_x$ emissions derived from JSEI, MEIC and DECSO in the Jiangsu section of the Yangtze River. MEIC does not include ship emissions and has a coarser resolution than the other inventories.

When compared to DECSO, $NO_x$ emissions from ships account for 6.1-74.5 % of the total emissions (Fig. 11). From Fig. 11, we see that the width of the Yangtze River varies from region to region. The width of the Yangtze River increases significantly
375     in Taizhou, Changzhou and the sections of the river close to the estuary of the Yangtze River (red box), where the river flow slows down. Because of the lower river speed, ship emissions in these areas might be overestimated.

As mentioned above, the JSEI $NO_x$ emissions contribute to about 28 % of total $NO_x$ emissions of MEIC in a region of 10-25 km around the river (Table 8). Compared to DECSO, the JSEI accounts for 42.9 % of total $NO_x$ emissions in the region of 5-10 km from the river. The MEIC grid cells cover a larger area than for DECSO and include more emissions than DECSO, so
380     the share of ship emissions for MEIC compared to DECSO is nearly half of that for DECSO. Even on the coarser grid of MEIC, the comparison shows that ship emissions from riverine areas account for at least 25 % of the total emissions, so ship emissions should not be ignored in $NO_x$ emission inventories.





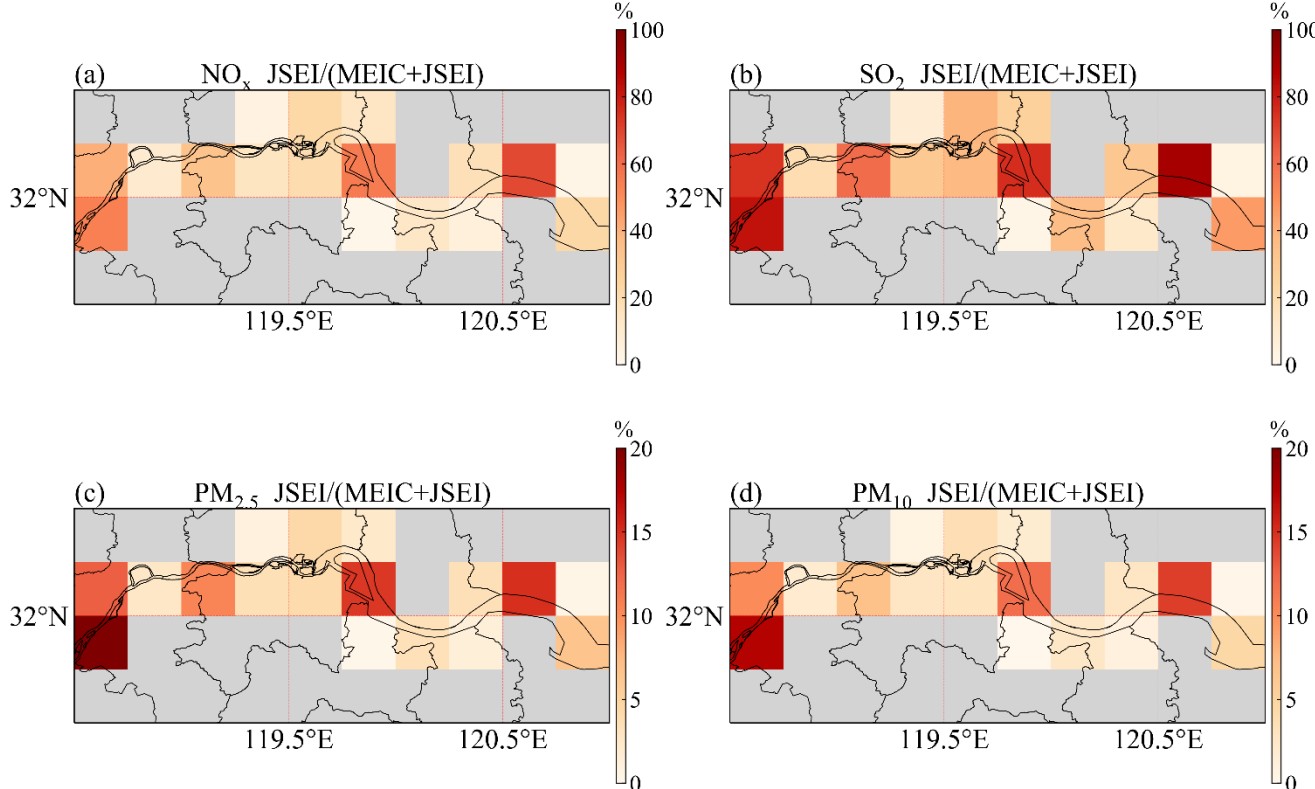

**Figure 9. Spatial distribution of the contribute of the river emissions (JSEI) to the total emissions (MEIC+JSEI) for NOₓ, SO₂, PM2.5 and PM10.**



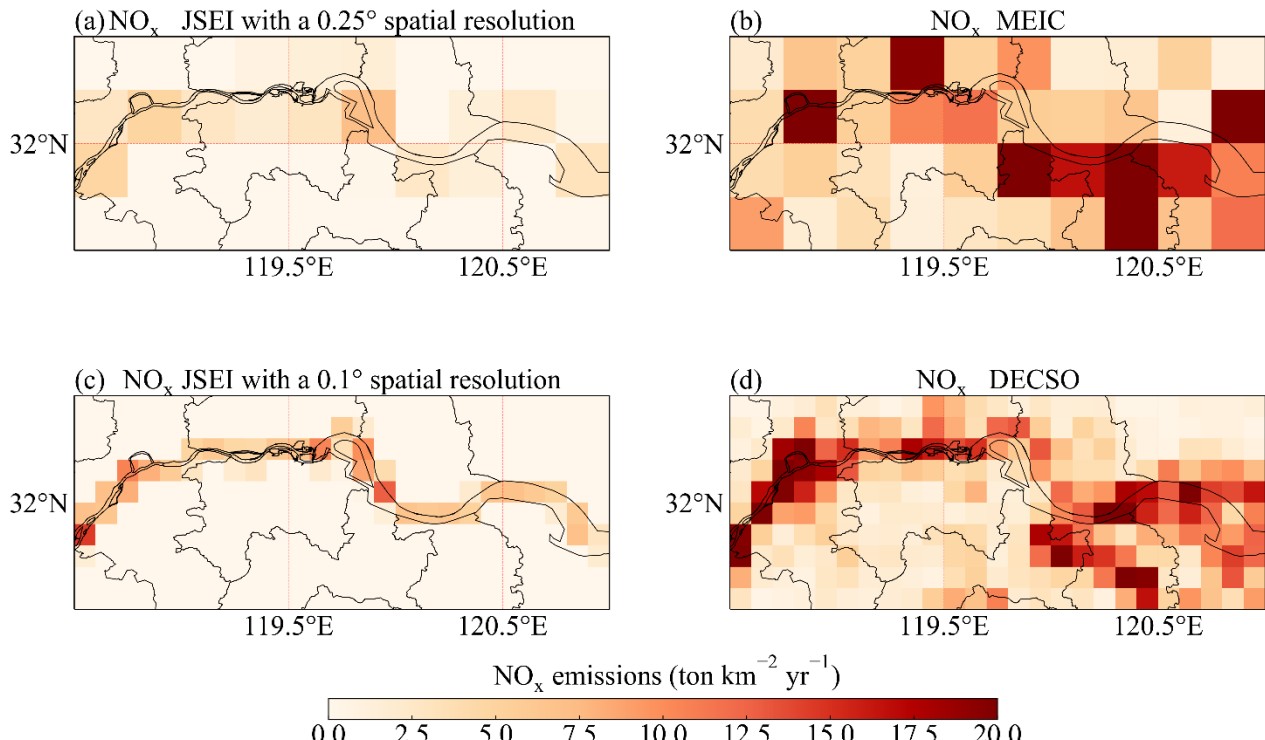

**Figure 10. Spatial distribution of NO$_x$ emissions for JSEI of 0.25° (a) and 0.1° (c) resolution, MEIC (b) and DECSO (d).**



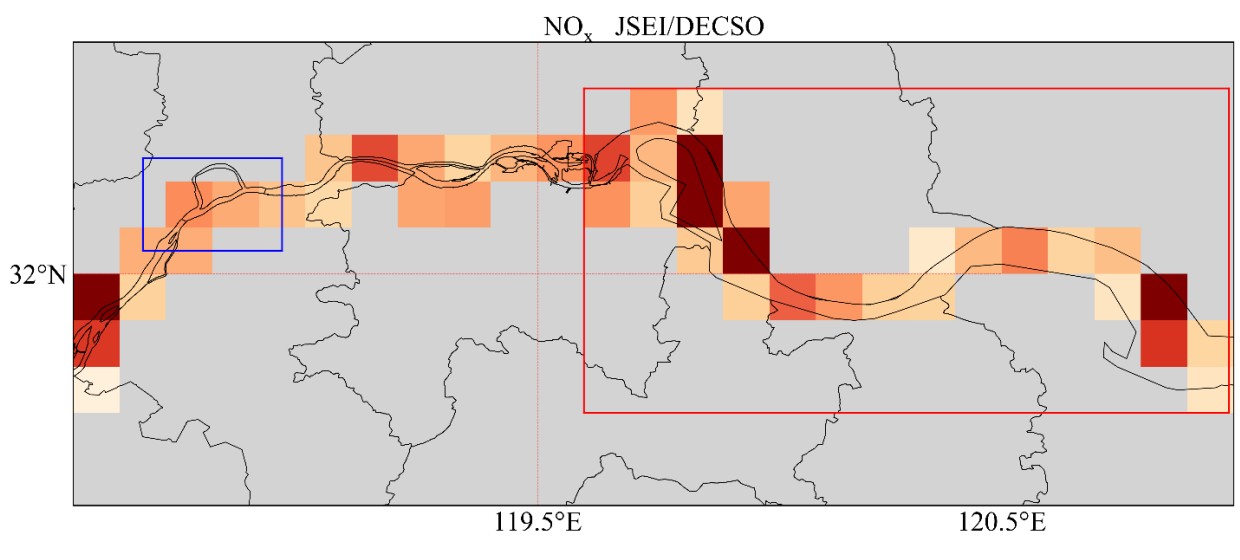

**Figure 11. Spatial distribution of the contribution of ship emissions (JSEI) to the total emissions (DECSO) for NO$_x$ on a 0.1 × 0.1 resolution. The blue box is the AIS observation area. The red box represents the section of the river with a broader width.**

Table 8. The total emissions in the selected river grid cells

|  | Year | Resolution | NO$_x$ (kton yr$^{-1}$) | Share (JSEI/Inventory) |
|---|---|---|---|---|
| **JSEI** | 2018 | 0.1 | 25.4 | 100 % |
| **MEIC** | 2017 | 0.25 | 91.3 | 27.8 % |
| **DECSO** | 2019 | 0.1 | 59.2 | 42.9 % |

**4.3 Comparison of the monthly emissions in the selected river grid cells**

Because SEIM is an annual emission inventory, we only compare the monthly variation of JSEI with MEIC and DECSO. Since the ship emissions for all species show similar monthly variability, we focus here only on NO$_x$. We see that the share of NO$_x$



emissions from ships in total NO$_x$ regardless of the inventory, is highest in the summer (Fig. 12). For the MEIC inventory, NO$_x$ emissions from ships can account for 17 % to 28 % of the NO$_x$ total emissions, and for DECSO, NO$_x$ emissions from ships account for 29 % to 57 % of the NO$_x$ total emissions.

The JSEI/(MEIC+JSEI) ratio tends to coincide with the monthly variation in ship emissions, as there is no significant monthly variation in emissions of the MEIC inventory. The JSEI/DECSO ratio shows that ship emissions are more important in summertime along the river. For DECSO, ship emissions accounted for more than 40 % of the total emissions in February, which shows that the pollution caused by ship activities during Spring Festival is quite significant in a time period of lower emissions in general.

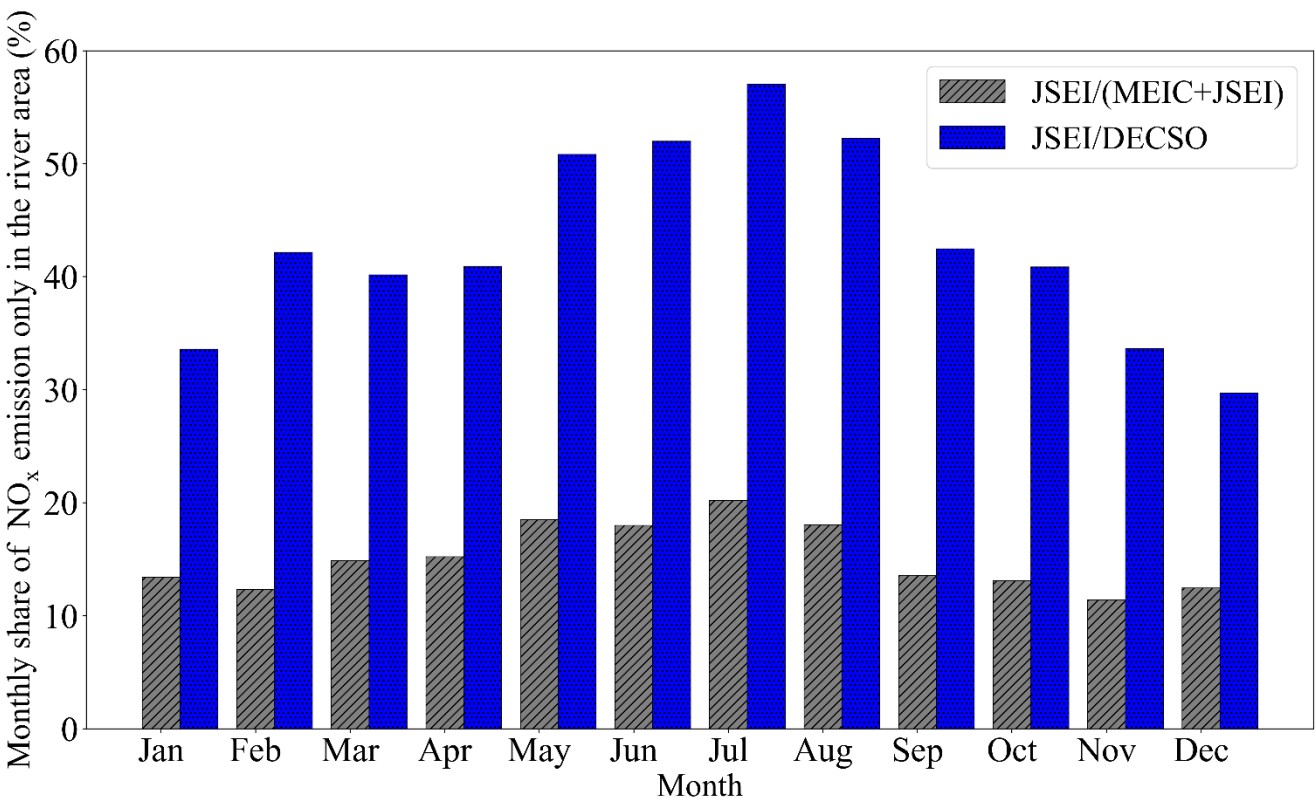

**Figure 12. Monthly contribution of NO$_x$ ship emissions to the total emissions (MEIC + JSEI, DECSO) around the river.**



## 5 Conclusions

Ship emissions are calculated per ship based on the real-time information reported by AIS. Since the AIS information is in general insufficient to support emission calculations, we propose a method to link the engine power and maximum speed to vessel type, length and speed to effectively supplement the missing ship data. In addition, we have presented a method using the river length per grid cell to extrapolate ship emissions along the river. Based on this method we have compiled a ship emission inventory with a resolution of 0.1° in the Jiangsu section of the Yangtze River.

The results show that the total ship emissions of $NO_x$, $SO_2$, $PM_{10}$, and $PM_{2.5}$ in the Jiangsu section of the Yangtze River from September 2018 to August 2019 were 83.5, 41.8, 3.8 and 3.3 kton, respectively. Cargo ships are the largest emitting ship type, followed by tankers and tugs. These three ship types accounted for 93 % of the total emissions. Tugs have the highest single-ship emissions. The monthly ship emissions are higher in the summer and lower in the winter, especially during the Spring Festival. Ship emissions are mainly concentrated in the main channel of the Yangtze River. Ship emissions in the side rivers are minimal, accounting for only 0.1 % of the total emissions, which is negligible.

Atmospheric emissions from ships are dominated by $NO_x$ and $SO_2$. Vessel traffic in the observation area is important because $NO_x$ emissions account for 46 % of the total $NO_x$ emissions compared to the DECSO inventory (grid of about 10 × 10 km). Our $NO_x$ results are similar to the SEIM inventory, up to 99 %, which shows the credibility of the results. Even if compared with the coarse resolution of the MEIC inventory (grid of about 25 × 25 km), the emissions from vessels along the Yangtze River (118.5°-121° E and 31.5°-32.5° N) account for approximately 28 % and 40 % of the total $NO_x$ and $SO_2$ emissions, respectively. Compared with the DECSO inventory based on satellite data, $NO_x$ emissions from ships along the Yangtze River account for 60 % of the total $NO_x$ emissions, which indicates that $NO_x$ along the river mainly comes from ship emissions, even higher than the emissions from the power plants or industry. Our study indicates that riverine ship emissions contribute significantly to air pollution. Riverine ship emissions can adversely affect the health of people living along the Yangtze River and have a negative impact on eco-system, biodiversity and eutrophication.

In the future, the ship emission inventory can be applied in a chemical transport model to explore in detail the impact of inland river ship emissions on air pollution of cities along the Yangtze River. The reliability of the air quality results can be verified

with observations from existing in situ stations or new Multi-Axis Differential Optical Absorption Spectroscopy (MAX-DOAS)

observations. This can provide a monitoring tool for policies such as the regulation of emission control standards for ships.

*Data availability.* The ship databases were obtained from the China Classification Society (CCS, https://www.ccs.org.cn/ccswz/, last access: 5 December 2022). The Multi-resolution Emission Inventory for China version 1.3
(MEIC v1.3) is available from http://meicmodel.org/ (last access: 5 December 2022). The Shipping Emission Inventory Model (SEIM) can be download from http://meicmodel.org.cn/?page_id=1770. Daily Emission estimates Constrained by Satellite Observations version 6.1 (DECSO v6.1) are published on https://www.temis.nl/emissions/data.php (last access: 5 December 2022).

*Author contributions.* RvdA and YY planed the campaign; Xiumei Zhang and RvdA performed the measurements; Xin Zhang
provided the ship data; Xiumei Zhang, RvdA and JD analyzed the data; Xiumei Zhang wrote the paper; all authors provided input on paper for revision before submission.

*Competing interests.* The authors declare that they have no conflict of interest.

*Acknowledgements.* We would like to thank the China Classification Society and Nanjing Maritime Bureau for their assistance. This research has received funding from the National Natural Science Foundation of China (grant no. 42075176) and the
program of China Scholarship Council (no. 202109040001).

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
