# Peer review of "Significant contribution of inland ships to the total $NO_x$ emissions along the Yangtze River"

_EGUsphere, 2022_

## Author Comment (AC1)

We thank the reviewers for their helpful comments. We have modified the manuscript as suggested. Below shows our responses to all the comments. The reviewer's comments will be shown in blue while our responses are in black, and changes made to the paper are shown in black block quotes. Unless otherwise indicated, page and line numbers correspond to the original manuscript.

The study of Zhang et al. built a ship emission inventory based on AIS signals and basic ship-related data for YRD region. The results were compared with several other emission inventories to verify reliability of the proposed approach and demonstrate the impact inland ship emissions on densely populated regions along the river.

The topic is interesting, but the publication has major deficiencies, is difficult to follow. In my view it cannot be published in the present form but will need substantial improvements.

Major issues:

1. The title is completely inappropriate. It's named 'The impact of inland ship emissions on air quality'. The title exaggerates the actual research scope seriously. The manuscript is almost entirely about emissions (Introduction, Methods and Discussion) rather than air quality. And the only description of air quality in Line 424-425 seems not correct, because the air pollution such as PM2.5 induced by ship emission also affected by other emission sources and meteorological condition. Therefore, the title should be modified at least. Otherwise, more quantitative study on air quality should be carried out. This study only investigated the Yangzi river- Jiangsu section, which is a short section, also can NOT represent YRD region as well. The title needs to honestly reflect the scope of the work done. The title is too big at present.

Thank you for noticing this issue. Yes, we agree that the title is not appropriate since we focus on the contribution of ship emissions to total emissions.

In lines 424-425 we link the ship $NO_x$ emissions to air pollution as $NO_2$ is one of the prime components of air pollution (for example reported by the WHO).

It has been specified in both the abstract and the text that the main study area is the Jiangsu section of the Yangtze River and does not represent the YRD region.

Based on the above considerations, we ultimately revised the title to "**Significant contribution of inland ships to the total $NO_x$ emissions along the Yangtze River**".

2. There are many mistakes in the review of the research status and the current situation of ship emission control. The authors need to re-write the literature review to include the most advanced and high quality research.

We thank the reviewer for the remark, but unfortunately the reviewer did not specify his/her concern with the introduction. We have checked the literature review again and added recent literature. The text has been modified.

In lines 35-41:

"… surrounding areas and increased traffic in the connected rivers. **Ship emissions in the YRD are much higher than those in the Bohai Bay and Pearl River Delta (PRD), reaching about 50% of the total emissions in these three regions (Chen et al., 2017; Wan et al., 2020).**

**Shipping emissions affecting air quality in the YRD region are mainly within 12 nautical miles of the coastline (Li et al., 2018). They can contribute between 30 % and 90 %, for example, over 75 % of ship-related $SO_2$ concentrations and 50 % of ship-related $PM_{2.5}$ concentrations (Lv et al., 2018; Feng et al., 2019). The data from the Ministry of Transport of China shows that by the end of 2021, the number of inland river transport vessels is 11.36 million vessels, which is higher than the sum of coastal transport and ocean transport in China (MOT, 2022). As one of the most economically developed regions in the east of China, the Yangtze River Delta (YRD) region is the busiest inland river ship transportation corridor in China.** Therefore, we focus on inland river ships…"

In line 54:

"…**Georgoulias et al. (2020) firstly combined observations from the TROPOspheric Monitoring Instrument onboard the Sentinel 5 Precursor satellite (TROPOMI/S5P) with AIS data to measure $NO_2$ plumes that could be detected and attributed to individual ships. In recent years, studies based on the method of combining satellite data with AIS data have been carried out mostly over seas (Kurchaba et al., 2022; Riess et al., 2022) but seldom over rivers.**"

In line 63:

"…**Li et al. (2018) and Wang et al. (2021) showed that the latest control policies have been effective in reducing ship emissions.** However, …

3.    The control policy of inland ships is different from that of oceangoing ships. Even for oceangoing ships, China's current control policy is not only in the three major control areas, but for the coastal waters of 12 nautical miles. The author has not quoted the latest research on inland river-related ship emissions.

We added the policies for 2016 to 2019 to the manuscript.

In lines 60-63:

"… because the Ministry of Transport in China started the emission control of ship emissions in 2016**(MOT, 2015). Domestic Emission Control Areas (DECAs) were set-up in the waters of Bohai Bay, YRD and PRD to control, for example the sulfur content of the fuel. In addition, equipment with AIS is becoming a mandatory requirement for ships of 100 gross ton and above. By the end of 2018, DECAs were expanded from the major ports to 12 nautical miles outside the coastline. The range of the Yangtze River covered by the DECAs was also raised from the original offshore cities to stretch from Shuifu in Yunnan to Liuhekou in Jiangsu (MOT, 2018). Li et al. (2018) and Wang et al. (2021) showed that the latest control policies have been effective in reducing ship emissions.** However, …"

4.    This study assumed that the average sulfur content of marine distillates (MD) of 1.5 % in the study area according to Xu et al. (2019), which was based on the data of 2014. However, this study was implemented from 2018 to 2019. The sulfur content for inland ships was totally different, which makes the results wrong.

Thanks for the advice. We have checked several studies and found the latest information.

Since 2016, authorities have been gradually limiting the sulfur content of marine fuels. Ordinary inland vessels typically use general diesel oil (GDO), while larger inland ships and direct river and sea vessels use Marine fuel oil (MFO). MFO includes residual oil (RO) and marine distillates (MD). The sulfur content requirements for different fuels are shown in Table S2.

For MFO sulfur content not higher than 0.001 % was scheduled to be implemented in January 2019, but actually in October 2018, DECAs started to response to this policy. Our study was carried in September 2018 to August 2019, the average fuel sulfur content should be well below 1.5 %. The emission factor of $NO_x$ is closely related to the age and rotation speed of the ship's engine, the emission factor of $SO_2$ directly depends on the sulfur content of the ship's fuel, and the emission factor of PM is more influenced by the sulfur content of the ship's fuel. We revised the fuel correction factor (Table 2). With the modification of fuel correction factor, the $NO_x$, $SO_2$, $PM_{2.5}$, $PM_{10}$ emitted from ships in Jiangsu section of Yangtze River from September 2018 to August 2019 were 83.5, 0.04, 0.006 and 0.005 kton, respectively. The corresponding text, figures, and tables have been revised in the manuscript.

Using the former average fuel sulfur content of 1.5 % can be assumed to be the emissions of $SO_2$ and PM from ships in the absence of policy regulations.

We added in lines 152-157:

"… **Ordinary inland vessels typically use general diesel oil (GDO), while larger inland ships and direct river and sea vessels use Marine fuel oil (MFO). MFO includes residual oil (RO) and marine distillates (MD). The sulfur content requirements for different fuels are shown in Table S2.** The emission factors mentioned above are obtained based on the assumption that marine fuel is heavy oil with a sulfur content of 2.7 % (EPA, 2009). **In October 2018, the DECAs began responding to a new policy that the sulfur content of inland marine fuel cannot be higher than 0.001 % in 2019 (Wang et al. 2021). In our study, the actual inland ship emissions are calculated based on the sulfur content of 0.001 %, for which a fuel correction factor is needed. In addition, we will calculate ship emissions for an average fuel sulfur content of 1.5 % (Xu et al.,2019) for a scenario where the new policy would be absent.** The fuel correction factor for each …"

**Table S2.** Evolvement of fuel sulfur content requirements in inland rivers.

| Fuel type | | Year | | | | |
|---|---|---|---|---|---|---|
| | | 2016 | 2017 | 2018 | 2019 | 2020 |
| **general diesel oil (GDO)** | | ≤0.035 % | ≤0.035 % | ≤0.005 % | ≤0.001 % | |
| **Marine fuel oil (MFO)** | residual oil (RO) | | 0.1-3.5 % | | ≤0.001 % | |
| | marine distillates (MD) | | 0.1-1.5 % | | | |

**Table 2.** Fuel correction factor $(f_F)$ used in this study.

| Fuel type | Sulfur content (%) | $NO_x$ | $SO_2$ | $PM_{2.5}$ | $PM_{10}$ |
|---|---|---|---|---|---|
| Unregulated Marine fuel | 1.5 % | 0.9 | 0.56 | 0.47 | 0.47 |
| Regulated Marine fuel | 0.001 % | 0.9 | 0.0005 | 0.0007 | 0.0007 |

5.  The author mentioned "We set up an AIS receiver in Nanjing University of Information Science and Technology (NUIST) to collect ship information including ship position, speed and heading, ship name, ship length and ship type". What's the power threshold for this receiver can receive from ships? What is the percentage of missing signals? With the distance getting farther, there are more missing signals. How to evaluate the emissions of lost ships? More details of the AIS receiver should be presented

Thanks for the detailed questions.

○  What's the power threshold for this receiver can receive from ships?

The frequency range of the ship's AIS signal is 161.975 MHz - 162.025 MHz and we use an antenna with a frequency range between 144 MHz and 163 MHz. The receiver is passive and the reception depend mainly on the distance of the ships and meteorological circumstances.

○  What is the percentage of missing signals? With the distance getting farther, there are more missing signals. How to evaluate the emissions of lost ships?

Indeed, as the distance gets farther and farther away, more and more signals are lost. However, as stated in the manuscript, the antenna is capable of receiving ship signals within at least a 50 km radius, but we only selected ship signals within a radius of less than 25 km. This is the area where even ships with weak AIS transmitters can be tracked under all conditions. In lines 97-100 we mentioned:

"The black dots in Fig. 1c are an example of the locations of ships according to the AIS signals received in a time interval of 4 hours. It shows that the antenna can receive signals within at least a 50 km radius. A region with a longitude of less than 118.95° E and a latitude of more than 32.05° N was selected as the area where all ships, including those with weak AIS transmitter can be tracked under all conditions."

6.  Ship emission is not only related to the length of the river. For example, in some major ports, the emission is large while the length may not high. The author should present more proofs to prove the proposed method that estimates the emission to larger regions based on the length of the river per grid cell. Such as plotting the relationship of emission and the length for the observational area or listing some references.

The extrapolation of our results to the wider region of Jiangsu is based on a few assumptions as mentioned in the manuscript. In our study, the ports were not taken into account, so the

emissions from ships moored in the ports were not considered in this study but were taken into account in the uncertainty analysis.

When moored ships are not considered, ship emissions can be closely related to river length. The spatial distribution of ship pollution in the Jiangsu section of the Yangtze River obtained according to this method is consistent with previous studies (Xu et al., 2019; Zhu et al., 2019).

7. It was not clear that how the 3-4% underestimation in Line 327 be calculated, and except for the emission from the auxiliary engine, the emission from the boiler was also not considered. How much did the boiler emission account for?

As stated in the manuscript, 17 % of the ships in the observation area are in dock every day, and the emissions from this part of the ships are not taken into account. Moreover, the ratio between auxiliary and main engines of ships is about 20-25. 17 % multiplied by the ratio gives an underestimation of emissions.

However, during the navigation of a ship, the main engine and auxiliary engine of the ship are working at the same time. For a moored ship, the main engine stops working. We have neglected the auxiliary engines because of the conflicting numbers in the literature. Based on the study of Weng et al. (2020) in the Yangtze estuary, we estimate the underestimated emissions to be about 12 % due to neglection of the auxiliary engine and boil emissions at berth and underway.

For inland vessels, boiler emissions are much lower and generally not considered.

According to the above discussion, we have revised lines 325 to 327 in the manuscript:

**"However, during the navigation of a ship, the main engine and auxiliary engine of the ship are working at the same time. For a moored ship, the main engine stops working. On average, 17 % of the ships in the observational area are in dock every day, and this part of the ship emissions have not taken into account, but the auxiliary engines are still working. Based on the study of Weng et al. (2020) in the Yangtze estuary, we estimate that our emissions show an underestimation of about 12 % because of ignoring the auxiliary engine and boiler emissions at berth and underway."**

8. Almost any sources of emission will account for a large share ratio in the grid where they emit. To illustrate the importance of emissions for inland vessels, the author should compare emissions in specific cities or regions (coastal cities, YRD …) rather than the selected river grid cells.

Thanks for the suggestion. Comparing to the selected river grid is important because these grid cells miss shipping emissions in the bottom-up inventories.

In addition, we have added the comparison for the Jiangsu province. Since the scope of our study was the Yangtze River section in Jiangsu Province, the total pollutant emissions from ships were compared with the total non-vessel anthropogenic emissions in Jiangsu Province in 2017. $SO_2$, $NO_x$, $PM_{10}$ and $PM_{2.5}$ from ships account for 0.01 %, 5 %, 0.001 % and 0.001 % of the total non-vessel anthropogenic $SO_2$, $NO_x$, $PM_{10}$ and $PM_{2.5}$ in Jiangsu Province, respectively, given by the Multi-resolution Emission Inventory of China (MEIC,

http://www.meicmodel.org/).

**Table S4.** Non-ship emissions given by MEIC and estimated ship emissions in the Jiangsu section of the Yangtze River (kton yr$^{-1}$)

| Sectors | SO$_2$ | NO$_x$ | PM$_{10}$ | PM$_{2.5}$ |
|---|---|---|---|---|
| Power | 46.6 | 318.6 | 65.6 | 39.7 |
| Industry | 235.3 | 649.4 | 268.2 | 174.6 |
| Residential | 8.8 | 24.4 | 115.7 | 111.2 |
| Transportation | 24.6 | 545.9 | 42.4 | 41.3 |
| **Total non-ship emissions** | **315.3** | **1538.3** | **491.9** | **366.8** |
| **Ship emissions** | **0.04** | **83.5** | **0.006** | **0.005** |

We added in line 382:

"… in NO$_x$ emission inventories. **When comparing the total ship emissions with the total non-vessel anthropogenic emissions in Jiangsu Province in 2017, NO$_x$ emissions from ships still account for 5 % of the total non-vessel anthropogenic NO$_x$ in Jiangsu Province given by the MEIC inventory (Table S4).**"

9.    The author should explain why there was a big difference between JESI and SEIM, especially for SO$_2$ and PM$_{2.5}$ in Section 4.1. Additionally, in the latest version of SEIM (Wang et al., 2021. Ship emissions around China under gradually promoted control policies from 2016 to 2019), the emissions over inland rivers are all included.

In the manuscript, we added an explanation of the reasons for the big differences between JSEI and SEIM. For the latest version of SEIM, the database is not yet publicly available. Here, we use SEIM v1.0.

In lines 358-359:

"… SEIM has higher values than JSEI for SO$_2$ and PM$_{2.5}$ **because SEIM calculated the emissions for 2017, when only major ports needed to strictly control the sulfur content of marine fuel. The sulfur content of marine fuel was 0.001 % in our study. In comparison, the sulfur content of ocean-going marine fuel in 2017 was about 2.7 %, much higher than that of inland river ship fuel. Ship pollutants that are greatly affected by the sulfur content of marine fuel, such as SO$_2$ and PM, will be reduced with the reduction of sulfur content. This shows that from 2017 to 2019, the policy was of a great significance for the ship emissions, effectively reducing the emissions of SO$_2$ and PM. For NO$_x$ emissions both inventories compare remarkably well**."

In line 244:

"… a total of seven species **(Liu et al., 2016, 2019).** However, emissions over inland rivers, except for the delta region, are not included. **Emissions from ocean-going vessels, coastal vessels and river vessels have been calculated in the latest version of SEIM (SEIM v2.0) (Wang et al., 2021). Here, we use SEIM v1.0, since only this version is publicly available.**"

Smaller issues:

1. Line 183, it should be Figure S1 rather than S2.

We have fixed the text.

"Figure **S1** shows fitted linear relationship between main engine power…"

2. In Figure 9 and Figure 11, why the sharing ratio of JESI is low in some grid of which most part is river?

There are many large factories along the river, such as the Nanjing Iron and Steel Group (https://en.wikipedia.org/wiki/Nanjing_Iron_and_Steel_Group), China Petrochemical Corporation (http://www.sinopecgroup.com/group/en/gywm/about.shtml) and Changzhou Xinbei Industrial Park. The pollution emitted by these factories is much larger than that emitted by ships. Therefore, the sharing ratio of JSEI will be small at these grids. For easier understanding, the locations of these plants are posted on the MEIC spatial distribution map.

[Figure]

**Figure R1.** Spatial distribution of NOx emissions for MEIC. The arrows point to the important industrial locations, the boxes show the name of the industrial complex, and the circle represents the area covered by the factories.

3. Line 379, why the author said "The MEIC grid cells cover a larger area than for DECSO and include more emissions than DECSO".

As stated in lines 377-379 of the text, the grid of MEIC is within the area of 10-25 km around the river and DECSO is within the area of 5-10 km from the river. Hence the grid cells of MEIC have higher total emissions than those of DECSO. To make it easier to understand, we describe it in a more direct way:

"**The MEIC grid cells are larger than the DECSO grid cells and therefore include more**

**emissions than for DECSO, so the share of ship emissions for MEIC is nearly half of that for DECSO.**"

4.   In Line 402, the author said "ship emissions accounted for more than 40 % of the total emissions in February, which shows that the pollution caused by ship activities during Spring Festival is quite significant in a time period of lower emissions in general". However, according to MEIC, the changes in NOx emission caused by the Spring Festival is not significant. Thus, which share is more reasonable, JESI/(JESI+SEMI) or JESI/DECSO. The author should give more proofs to support this sentence.

Our conclusion is based on the following.

○   As stated in issue 3, a MEIC grid cell has a larger area and with same ship emissions, the effect on the ratio would be much smaller than for JSEI/DECSO. In this case, the JSEI/DECSO ratio can give us more information on the share of ship emissions in total emissions.

○   In terms of monthly variability of the MEIC and DECSO inventories in the river region only, MEIC was relatively constant throughout the year, with no particularly high or low values. This is also mentioned in line 399-400 of the text:

"The JSEI/(MEIC+JSEI) ratio tends to coincide with the monthly variation in ship emissions, as there is no significant monthly variation in emissions of the MEIC inventory."

[Figure]

**Figure R2.** Seasonal cycle of NO$_x$ emissions in the river area only for JSEI, MEIC and DECSO.

○   The temporal resolution of the officially released version of SEIM is annual, and therefore, it is impossible to analyze its monthly changes.

5. The format of references should be unified. (Such as in Line 237)

Thanks for pointing this out. In line 237, the format of references is unified.

[revised manuscript text omitted]

---

## Author Comment (AC2)

We thank the reviewers for their helpful comments. We have modified the manuscript as suggested. Below shows our responses to all the comments. The reviewer's comments will be shown in blue while our responses are in black, and changes made to the paper are shown in black block quotes. Unless otherwise indicated, page and line numbers correspond to the original manuscript.

In this paper, the authors present ship emission calculations for the Yangtze River area by combining AIS ship location and other data from the China Classification Society (CCS). By using simple parameterizations based on the specific ship characteristics (type, engine, length, speed, etc.) they compile a monthly inventory for a section of the Yangtze River and compare it with an annual inventory (SEIM) and two monthly inventories (DECSO and MEIC). I enjoyed reading the paper, as it is a significant contribution towards understanding the significance of inland ship emissions on the $NO_x$, $SO_2$ and PM levels in the broader areas around rivers. The methodology has also potential of being used in other areas (e.g. central Europe) with significant ship traffic across densely populated areas. To this end the paper deserves to be published but has to be improved before. See my comments below:

1. The title is not appropriate. The authors should use a title describing the exact focus of their study, e.g. "Inland ship emissions across the Yangtze River area in China" or "Inland ship emissions: the case of Yangtze River area in China", etc.

Thank you for the helpful suggestion. Yes, we agree that the title is not appropriate and we have revised the title to "**Significant contribution of inland ships to the total $NO_x$ emissions along the Yangtze River**". See also our response to reviewer 1.

2. The paper should be revised as the flow is not very pleasant and there are several minor expression mistakes. There are many very small phrases. E.g. "…Note the limitations of our method. Some ships would reduce their speed when going downstream…" which does not help the reader at all.

Thank you for the comments. We have carefully checked the manuscript and made several revisions throughout the text.

3. The authors should consider discussing new developments in ship emission detection and attribution in their introduction. For example, it has been shown that today the AIS data combined with satellite data can give as an indication of the pollution produced by individual ships (see Georgoulias et al., 2020; Riess et al., 2022; Kurchaba et al., 2022). The use of ground-based DOAS methods to infer individual ship emissions from rivers and channels is also possible as shown in Krause et al. (2021).

Thanks for the suggestion. We have added these and several other references to the introduction and discussion.

In lines 35-41:

"… surrounding areas and increased traffic in the connected rivers. **Ship emissions in the YRD are much higher than those in the Bohai Bay and Pearl River Delta (PRD),**

reaching about 50% of the total emissions in these three regions (Chen et al., 2017; Wan et al., 2020). Shipping emissions affecting air quality in the YRD region are mainly within 12 nautical miles of the coastline (Li et al., 2018). They can contribute between 30 % and 90 %, for example, over 75 % of ship-related $SO_2$ concentrations and 50 % of ship-related $PM_{2.5}$ concentrations (Lv et al., 2018; Feng et al., 2019). The data from the Ministry of Transport of China shows that by the end of 2021, the number of inland river transport vessels is 11.36 million vessels, which is higher than the sum of coastal transport and ocean transport in China (MOT, 2022). As one of the most economically developed regions in the east of China, the Yangtze River Delta (YRD) region is the busiest inland river ship transportation corridor in China. Therefore, we focus on inland river ships…"

In line 54:

"…Georgoulias et al. (2020) firstly combined observations from the TROPOspheric Monitoring Instrument onboard the Sentinel 5 Precursor satellite (TROPOMI/S5P) with AIS data to measure $NO_2$ plumes that could be detected and attributed to individual ships. In recent years, studies based on the method of combining satellite data with AIS data have been carried out mostly over seas (Kurchaba et al., 2022; Riess et al., 2022) but seldom over rivers."

In line 431:

"… or ground-based Differential Optical Absorption Spectroscopy (DOAS) observations along the river (Cheng et al., 2019; Krause et al., 2021). "

4.      When discussing the method and specifically the engine power (EP) calculations, it is not very clear how the authors calculate the related regression between EP and L^2 x v^3. Probably the authors used EP data from CCS to do the regression; however, in the beginning of the paragraph they write "Since the engine power is missing in the AIS data, we develop a method to relate the engine power to the ship type, length and speed. Those parameters are available in the AIS data." Are EP data available only for a fraction of the ships? Please refine this!

The AIS data provides real-time information such as vessel position, speed and heading, as well as static vessel information such as vessel name, vessel length and vessel type. Thus, for each ship, we can get its type, length and speed from the AIS data, but the ship main engine power is missing.

The CCS database provides data such as ship type, main engine power, maximum ship designed speed, ship length and year of ship built. Using the CCS database, we can derive the regression relationship for each category of ship by linear fitting of this proxy( $EP \sim L^2 \times V_{\max}^3$ ).

We clarified this by modifying the text. In lines 169-170:

"Since the engine power is missing in the AIS data, we develop a method to relate the engine power to the ship type, length and speed. These parameters are available in the AIS data unlike the engine power."

And in lines 179-182:

"… with its length and its speed as: $P \sim L_s^2 v^3$. **The China Classification Society (CCS, https://www.ccs.org.cn/ccswz/, last access: 27 February 2023) database of Chinese domestic ships provides data such as ship type, main engine power, maximum ship designed speed, ship length and construction year of the ship. Using these ship parameters, we can derive the average regression relationship for each category of ship by linear fitting of this proxy ($P \sim L_s^2 v^3$).** The fitted linear relation …"

5.  On top of my previous comment, the authors might include in Table 3 apart from the slope the whole equation (slope + intercept + the corresponding uncertainties). The uncertainties induced by the use of this equations might be incorporated into the discussion in paragraph 3.4 where the authors discuss only two sources of uncertainty.

In the manuscript we fitted the intercept, but the fitted intercept was negligible and therefore not included. Specifically, you can see Figure S1 in the supplement.

For this method, it is equivalent to averaging over each ship type. When actually using this linear relationship, the main engine power of some ships will be on the high or low side. But in the overall perspective, it is not a big difference.

For the uncertainties of ship emissions, the contribution of this part is included but very limited, and other sources of uncertainty are described in the answer to the next question.

6.  The uncertainty discussion is very limited. Please discuss more aspects or integrate this in another paragraph.

We have added an extra section of the uncertainties.

3.4 Uncertainty

"**In this section we will discuss the uncertainty on our emission inventory. Our calculations have been based on the main engine only. However, during the navigation of a ship, the main engine and auxiliary engine of the ship are working at the same time. For a moored ship, the main engine stops working. On average, 17 % of the ships in the observational area are in dock every day, and this part of the ship emissions have not taken into account, but the auxiliary engines are still working. Based on the study of Weng et al. (2020) in the Yangtze estuary, we estimate that our emissions show an underestimation of about 12 % because of ignoring the auxiliary engine and boiler emissions at berth and underway.**

**However, the locations of high ship emissions are consistent with previous studies. Zhu et al. (2019) pointed out that the distribution of ship emissions in the Jiangsu section of the Yangtze River in 2017 was uneven, with the emission rates in the Nanjing section of the Yangtze River and the Jiangyin section of the Yangtze River being relatively high. Xu et al. (2019) noted that for ports along the river, Nanjing port had the highest rate of ship emissions.**

**As the Yangtze River becomes wider when getting closer to the sea, the speed of the river will be reduced and thus the emissions from ships can be lower. In the extreme**

**cases of stagnant water, ship emissions can be reduced by a maximum of 3-33 % depending on the month. We estimate that this may lead to an overestimation of about 10 % in the ship emissions outside our study area around Nanjing.**

**Currently, the AIS-based approach is considered as the best practice for ship inventories. However, there is still a lack of reliable local emission factors, auxiliary engine power ratings and fuel correction factor in the YRD region, which contributes largely to the uncertainties in this study. The selection of accurate emission factors is critical to the calculation of the ship emission inventory and the uncertainty that comes with it. The emission factors are closely related to the age and rotation of the ship's engine as well as the engine load, and the fuel correction factor depends on the sulfur content of the marine fuel. Earlier heavy oil was a fuel of low quality with a sulfur content of about 2.7 %. In contrast, the fuel sulfur content in this study is only 0.001 %, while in the beginning of time period the fuel sulfur content may be as high as 1.5 %. For the scenario that the sulfur content is not regulated, we have calculated that the SO2 and PM emissions would be about a factor 700-1000 higher.**

**In conclusion, our derived emissions have an underestimation of 12 % due to ignoring the auxiliary engines and boilers and an overestimation in some regions of about 10 % due to the slower river flow. Adding this to the uncertainties in emission factors we estimate the total uncertainty to be 5-15 %.**"

7.    The title of section 4 should be changed. E.g. "Importance of inland ship emissions" might be "Contribution of inland ship emissions relative to emissions from other sources".

We agree with this suggestion. Now the title of section 4 is "**Contribution of inland ship emissions relative to emissions from other sources**".

8.    Please comment on the significant difference between JSEI and SEIM compared to the other two inventories.

The differences between the ship emission inventories (JSEI, SEIM) compared to MEIC and DECSO are illustrated below in two aspects. On the one hand, the differences between the two ship emission inventories themselves, and on the other hand, the comparison of the ship emission inventories to the total emission inventories (MEIC, DECSO).

○    Difference between JSEI and SEIM

Although both JSEI and SEIM are ship emission inventories, the difference in base year and ship type leads to some differences between them. In lines 355-359:

"When comparing the overlapping grid cells, JSEI accounts for in average about 99 %, 0.05 % and 0.06 % of the SEIM emissions for $NO_x$, $SO_2$ and $PM_{2.5}$, respectively. The average emissions from inland ships over rivers (JSEI) compare well with average emissions of sea-going ships (SEIM) for $NO_x$. SEIM has higher values than JSEI for $SO_2$ and $PM_{2.5}$ **because SEIM calculated the emissions for 2017, when only major ports**

**needed to strictly control the sulfur content of marine fuel. The sulfur content of marine fuel was 0.001 % in our study. In comparison, the sulfur content of ocean-going marine fuel in 2017 was about 2.7 %, much higher than that of inland river ship fuel. Ship pollutants that are greatly affected by the sulfur content of marine fuel, such as $SO_2$ and PM, will be reduced with the reduction of sulfur content. This shows that from 2017 to 2019, the policy was of a great significance for the ship emissions, effectively reducing the emissions of $SO_2$ and PM. For $NO_x$ emissions both inventories compare remarkably well**."

○ Compare to MEIC and DECSO

The current publicly available version of SEIM with a resolution of year has the same base year as MEIC, both in 2017. SEIM v1.0 has fewer grid cells in inland rivers and could not represent the contribution of inland river vessel pollution emissions to total emissions when compared to the total emissions. Table R1 shows the contribution of ship emissions to the total $NO_x$, $SO_2$ and $PM_{2.5}$ emissions in the Jiangsu section of the Yangtze River (118.5°-121° E and 31.5°-32.5° N). In the inland segments where the SEIM inventory is missing, $NO_x$ emissions can reach about 7-10 % of the total $NO_x$ emissions, both compared to MEIC and DECSO. So inland ship $NO_x$ emissions cannot be ignored.

**Table R1.** Contribution of ship emissions to the total $NO_x$, $SO_2$ and $PM_{2.5}$ emissions in the Jiangsu section of the Yangtze River (118.5°-121° E and 31.5°-32.5° N).

|  | SEIM | | JSEI | |
| --- | --- | --- | --- | --- |
|  | SEIM/(SEIM+MEIC) | SEIM/DECSO | JSEI/(JSEI+MEIC) | JSEI/DECSO |
| **$NO_x$** | 3.3 % | 4.5 % | 10 % | 14 % |
| **$SO_2$** | 13 % |  | 0.14 % |  |
| **$PM_{2.5}$** | 1.5 % |  | 0.01 % |  |

9. In Fig 12, the authors might add in the legend the resolution of the models to make clear that the difference in the bars is mostly related to the different resolutions.

We have replotted Figure 12 to make it easier for the readers to understand the difference in the bars due to the different resolutions.

[Figure]

**Figure 12.** Monthly contribution of NO$_x$ ship emissions to the total emissions (MEIC + JSEI, DECSO) for the grid cells including the river. The resolution of JSEI/(MEIC + JSEI) is 0.25 °, the resolution of JSEI/(DECSO) is 0.1 °.

**References**

Chen, D., Wang, X., Nelson, P., Li, Y., Zhao, N., Zhao, Y., Lang, J., Zhou, Y., and Guo, X.: Ship emission inventory and its impact on the $PM_{2.5}$ air pollution in Qingdao Port, North China, Atmos. Environ., 166, 351–361, https://doi.org/10.1016/j.atmosenv.2017.07.021, 2017.

Cheng, Y., Wang, S., Zhu, J., Guo, Y., Zhang, R., Liu, Y., Zhang, Y., Yu, Q., Ma, W., and Zhou, B.: Surveillance of $SO_2$ and $NO_2$ from ship emissions by MAX-DOAS measurements and the implications regarding fuel sulfur content compliance, Atmos. Chem. Phys., 19, 13611–13626, https://doi.org/10.5194/acp-19-13611-2019, 2019.

Feng, J., Zhang, Y., Li, S., Mao, J., Patton, A. P., Zhou, Y., Ma, W., Liu, C., Kan, H., Huang, C., An, J., Li, L., Shen, Y., Fu, Q., Wang, X., Liu, J., Wang, S., Ding, D., Cheng, J., Ge, W., Zhu, H., and Walker, K.: The influence of spatiality on shipping emissions, air quality and potential human exposure in the Yangtze River Delta/Shanghai, China, Atmos. Chem. Phys., 19, 6167–6183, https://doi.org/10.5194/acp-19-6167-2019, 2019.

Georgoulias, A. K., Boersma, K. F., van Vliet, J., Zhang, X., van der A, R., Zanis, P., and de Laat, J.: Detection of $NO_2$ pollution plumes from individual ships with the TROPOMI/S5P satellite sensor, Environ. Res. Lett., 15, 124037, https://doi.org/10.1088/1748-9326/abc445, 2020.

Krause, K., Wittrock, F., Richter, A., Schmitt, S., Pöhler, D., Weigelt, A., and Burrows, J. P.: Estimation of ship emission rates at a major shipping lane by long-path DOAS measurements, Atmos. Meas. Tech., 14, 5791–5807, https://doi.org/10.5194/amt-14-5791-2021, 2021.

Kurchaba, S., van Vliet, J., Verbeek, F. J., Meulman, J. J., and Veenman, C. J.: Supervised Segmentation of $NO_2$ Plumes from Individual Ships Using TROPOMI Satellite Data, Remote Sens., 14, 5809, https://doi.org/10.3390/rs14225809, 2022.

Li, C., Borken-Kleefeld, J., Zheng, J., Yuan, Z., Ou, J., Li, Y., Wang, Y., and Xu, Y.: Decadal evolution of ship emissions in China from 2004 to 2013 by using an integrated AIS-based approach and projection to 2040, Atmos. Chem. Phys., 18, 6075–6093, https://doi.org/10.5194/acp-18-6075-2018, 2018.

Lv, Z., Liu, H., Ying, Q., Fu, M., Meng, Z., Wang, Y., Wei, W., Gong, H., and He, K.: Impacts of shipping emissions on $PM_{2.5}$ pollution in China, Atmos. Chem. Phys., 18, 15811–15824, https://doi.org/10.5194/acp-18-15811-2018, 2018.

Ministry of Transport (MOT): Statistical bulletin on the development of the transport sector in 2021, available at: https://xxgk.mot.gov.cn/2020/jigou/zhghs/202205/t20220524_3656659.html (last access: 27 February 2023), 2022.

Riess, T. C. V. W., Boersma, K. F., van Vliet, J., Peters, W., Sneep, M., Eskes, H., and van Geffen, J.: Improved monitoring of shipping $NO_2$ with TROPOMI: decreasing $NO_x$ emissions in European seas during the COVID-19 pandemic, Atmos. Meas. Tech., 15, 1415–1438, https://doi.org/10.5194/amt-15-1415-2022, 2022.

Wan, Z., Ji, S., Liu, Y., Zhang, Q., Chen, J., and Wang, Q.: Shipping emission inventories in China's Bohai Bay, Yangtze River Delta, and Pearl River Delta in 2018, Mar. Pollut. Bull., 151, 110882, https://doi.org/10.1016/j.marpolbul.2019.110882, 2020.

---

## Author Response (AR2)

We thank the reviewers for their helpful comments. We have modified the manuscript as suggested. Below shows our responses to all the comments. The reviewer's comments will be shown in blue while our responses are in black, and changes made to the paper are shown in black block quotes. Unless otherwise indicated, page and line numbers correspond to the original manuscript.

**For reviewer 1:**

The revised version has shown significant improvement, and all my concerns regarding the previous version have been appropriately and clearly addressed. Only a few minor comments to be considered. There's no need for a further round of review.

Firstly, it would be beneficial for the author to include a brief overview of the current state of ship emission control and its associated air quality impact for inland vessels. This inclusion will highlight the importance of this study beyond just ships operating within 12 or 200 Nm.

Thanks for the advice. We added some discussions about control policies in lines 70-71 :

> "**Wang et al. (2021) presented a detailed time line of the control policies, with the DECAs 1.0 policy starting in 2017 and DECAs 2.0 in 2019, while for river vessels the fuel sulfur content had to be gradually reduced to 10 ppm in the last half year of 2017.** Wang et al. (2021) showed that the latest control policies have been effective in reducing ship emissions **of $SO_2$ and PM, especially after 2018, when the DECAs 2.0 control policy became effective. The scenario study of the DECAs policies (Li et al., 2018) showed that several decades will be taken to reach a similar reduction for $NO_x$ emissions compared to $SO_2$ and PM, and this requires significant technological changes for ships…**"

Secondly, it is suggested to incorporate Section 4.1, "A comparison of the ship emission inventories," into Section 3. The comparison between JSEI and SEIM does not align with the theme of Section 4 and would be better suited within Section 3.

Thanks for the suggestion. The subject of section 4 is the contribution of inland ship emissions relative to emissions from other sources. Section 4.1 emphasizes the importance of inland ship emissions compared with sea-going ship emissions. Following this thought, the contribution of inland vessel emissions to total emissions is analyzed (section 4.2 & 4.3).

**References**

Li, C., Borken-Kleefeld, J., Zheng, J., Yuan, Z., Ou, J., Li, Y., Wang, Y., and Xu, Y.: Decadal evolution of ship emissions in China from 2004 to 2013 by using an integrated AIS-based approach and projection to 2040, Atmos. Chem. Phys., 18, 6075–6093, https://doi.org/10.5194/acp-18-6075-2018, 2018.

Wang, X., Yi, W., Lv, Z., Deng, F., Zheng, S., Xu, H., Zhao, J., Liu, H., and He, K.: Ship emissions around China under gradually promoted control policies from 2016 to 2019, Atmos. Chem. Phys., 21, 13835–13853, https://doi.org/10.5194/acp-21-13835-2021, 2021.